# Integrative proteomic profiling of ovarian cancer cell lines reveals precursor cell associated proteins and functional status

F. Coscia[1,*], K.M. Watters[2,*], M. Curtis[2], M.A. Eckert[2], C.Y. Chiang[2], S. Tyanova[1], A. Montag[3], R.R. Lastra[3], E. Lengyel[2,**] & M. Mann[1,**]

A cell line representative of human high-grade serous ovarian cancer (HGSOC) should not only resemble its tumour of origin at the molecular level, but also demonstrate functional utility in pre-clinical investigations. Here, we report the integrated proteomic analysis of 26 ovarian cancer cell lines, HGSOC tumours, immortalized ovarian surface epithelial cells and fallopian tube epithelial cells via a single-run mass spectrometric workflow. The in-depth quantification of >10,000 proteins results in three distinct cell line categories: epithelial (group I), clear cell (group II) and mesenchymal (group III). We identify a 67-protein cell line signature, which separates our entire proteomic data set, as well as a confirmatory publicly available CPTAC/TCGA tumour proteome data set, into a predominantly epithelial and mesenchymal HGSOC tumour cluster. This proteomics-based epithelial/mesenchymal stratification of cell lines and human tumours indicates a possible origin of HGSOC either from the fallopian tube or from the ovarian surface epithelium.

[1] Department of Proteomics and Signal Transduction, Max Planck Institute of Biochemistry, 82152 Martinsried, Germany. [2] Department of Obstetrics and Gynecology, Section of Gynecologic Oncology, University of Chicago, Chicago, Illinois 60637, USA. [3] Department of Pathology, University of Chicago Medicine, Chicago, Illinois 60637, USA. * These authors contributed equally to this work. ** These authors jointly supervised this work. Correspondence and requests for materials should be addressed to E.L. (email: elengyel@uchicago.edu) or to M.M. (email: mmann@biochem.mpg.de).

nvasive ovarian cancer (OvCa) is a highly heterogeneous disease divided into four major histologic subtypes, namely serous, endometrioid, mucinous and clear cell OvCa. High-grade serous ovarian cancer (HGSOC) is the most common (70%) and aggressive subtype and is primarily responsible for the low survival rate[1]. Until recently, HGSOC was thought to originate exclusively in the ovaries, as tumours almost invariably involve the ovary. However, the discovery of a possible precursor lesion, serous tubal intraepithelial carcinoma (STIC), in the fallopian tube fimbria of BRCA-mutation carriers, as well as in HGSOC patients, provides strong evidence for the fallopian tube fimbria as the probable site of origin[2–5]. HGSOC is characterized by ubiquitous somatic TP53 mutations[6] and genetic instability[7], and frequently evolves to a chemo-resistant state. From a molecular standpoint, it is classified into sub-groups based on characteristic gene expression signatures–differentiated, immunoreactive, proliferative and mesenchymal[7,8].

Given that it is difficult to perform mechanistic studies with primary tissue, the necessity for cellular models for *in vitro* and *in vivo* experiments is apparent. However, these models should be as representative of the tumour as possible, as there is little clinical utility for experimental data obtained in cell lines that do not reflect the disease being studied; these results might be, at best, misleading and, at worst, harmful to patients. The time that has elapsed since many OvCa cell lines were established (some were created more than 30 years ago), coupled with the risk of switching or cross-contamination when propagated for a long time, and the only recent introduction of genomic 'fingerprinting' techniques, has led to the incorrect assignment of the tissue origin of many OvCa cell lines[9]. The recent establishment of The Cancer Genome Atlas (TCGA)[7] has opened the door for researchers to begin to address these uncertainties and should allow selection of the most representative cell lines on the basis of genomic and transcriptomic information. A number of recent studies have integrated these HGSOC genomic characteristics into their assessment of suitable cell lines, to better understand OvCa biology and find novel treatment targets. Domcke *et al.*[9] evaluated data from the Cancer Cell Line Encyclopedia (CCLE)[10] by comparing a panel of 47 OvCa cell lines to HGSOC tissue data available through the TCGA consortium[7] by means of copy number alterations (CNA), mutation frequency and gene expression data. The authors cautioned against the use of some of the more commonly used OvCa cell lines due to their poor overall resemblance to HGSOC in patients at the genomic level. Mitra *et al.*[11] and Elias *et al.*[12] recently highlighted some of the limitations of the newly described HGSOC cell lines in pre-clinical studies, including their limited and inconsistent ability to form tumours in immuno-deficient mice. Interestingly, the cells with less genetic resemblance to the TCGA tumours had a metastatic pattern that was very similar to that of human HGSOC (for example, disseminated abdominal tumour nodules and omental involvement, but no extra-abdominal metastasis)[11,12].

It is as yet unknown to what extent the molecular characteristics reported by all these studies are represented at the protein level. Genomic and transcriptomic level analyses do not necessarily reflect the phenotype-defining proteomic profile. So far, there has been no direct and in-depth proteomic comparison between OvCa cellular models and tumour tissues. However, given that proteins represent the functional and phenotype-defining units of a cell, a quantitative proteomics approach should be superior to gene expression-driven comparisons alone. We therefore hypothesized that an integrated and streamlined mass-spectrometry (MS)-based proteomics approach[13–15] is a promising methodology for molecular subtype characterization. We here demonstrate that directly integrating proteomic profiles from cell lines, tumour tissues and primary cells adds a highly informative level to the evaluation of OvCa cellular model systems. A cell line-derived 67-protein signature classifies OvCa tumours potentially arising in the ovarian surface epithelium (OSE) or fallopian tube epithelial cells (FTECs) and predicts functional properties of these cells. We also provide a user-friendly resource of the quantitative protein expression of 30 ovarian cell lines.

## Results

**Deep single-run proteomics of OvCa tissues and cell lines**. To the extent that proteomics has reached a reasonable depth of quantification, this has usually involved multi-step workflows with extensive fractionation and correspondingly long measurement times. Since we sought to compare a large number of OvCa proteomes, we instead adapted and applied a recently described method based on a single-run workflow[15,16]. Briefly, we performed tryptic digestion of the entire proteome followed by chromatographic separation using relatively long (4 h) HPLC gradients coupled to online mass spectrometric analysis on high-resolution quadrupole Orbitrap mass spectrometers (see the 'Methods' section). Using this workflow, we quantified the proteomes of eight HGSOC tumour tissues, 30 cell lines (26 OvCa; two cervical cancer; two immortalized ovarian surface epithelial (IOSE)) and three primary FTEC isolates (Fig. 1a).

In total, our analysis resulted in 229,004 unique peptide sequences corresponding to 11,070 distinct protein groups at a peptide and protein false-discovery rate (FDR) of less than 1% (refs 17,18; see the 'Methods' section). This remarkable depth, considering the absence of any fractionation step, was accompanied by a median protein sequence coverage of 43.6% by the identified peptides. Label-free protein quantification (LFQ) using the MaxLFQ algorithm[19] resulted in a median depth of 7,828 protein quantifications per single measurement. The LFQ values were highly reproducible as cell line replicates had median Pearson correlation coefficients of 0.95 (Supplementary Fig. 1a), whereas comparison of the two different IOSE cell lines and FTECs from two different healthy donors had only somewhat lower correlations of 0.93 each. Interestingly, HGSOC tumours from the left and right ovaries from a single patient featured a very high correlation of 0.95 at the proteome level (see also below; Fig. 1b). These results demonstrate high experimental reproducibility and cell-type homogeneity.

When we required quantification in at least two of three cell line replicates, we obtained a filtered data set of, in total, 10,926 proteins containing an average of 7,810 protein quantifications per cell line, 8,143 per HGSOC tumour tissue and 7,609 for the primary FTEC replicates (Fig. 1c). A total of 8,397 were present in all three filtered data sets (77%; Fig. 1d). More than 99% of the FTEC proteins were also detected in OvCa cell lines or tumour tissues and, likewise, nearly all proteins detectable in the tumour tissue at our depth of analysis were also found in the cell line proteomes (97%). The few unique proteins in the tumour data set were enriched for plasma proteins and these were filtered out in subsequent analyses (see the 'Methods' section). There was even a high representation of the OSE proteome as represented by IOSE cell lines in the combined cancer cell line proteome (99%) and a very high overlap with the FTECs (91%). Furthermore, protein intensities were comparably distributed between these systems, as were the percentages of total protein mass attributable to major cellular compartments (Supplementary Fig. 1b, Supplementary Fig. 1c).

**Protein expression is heterogeneous in OvCa cell lines**. We first analysed commonalities and differences in protein expression

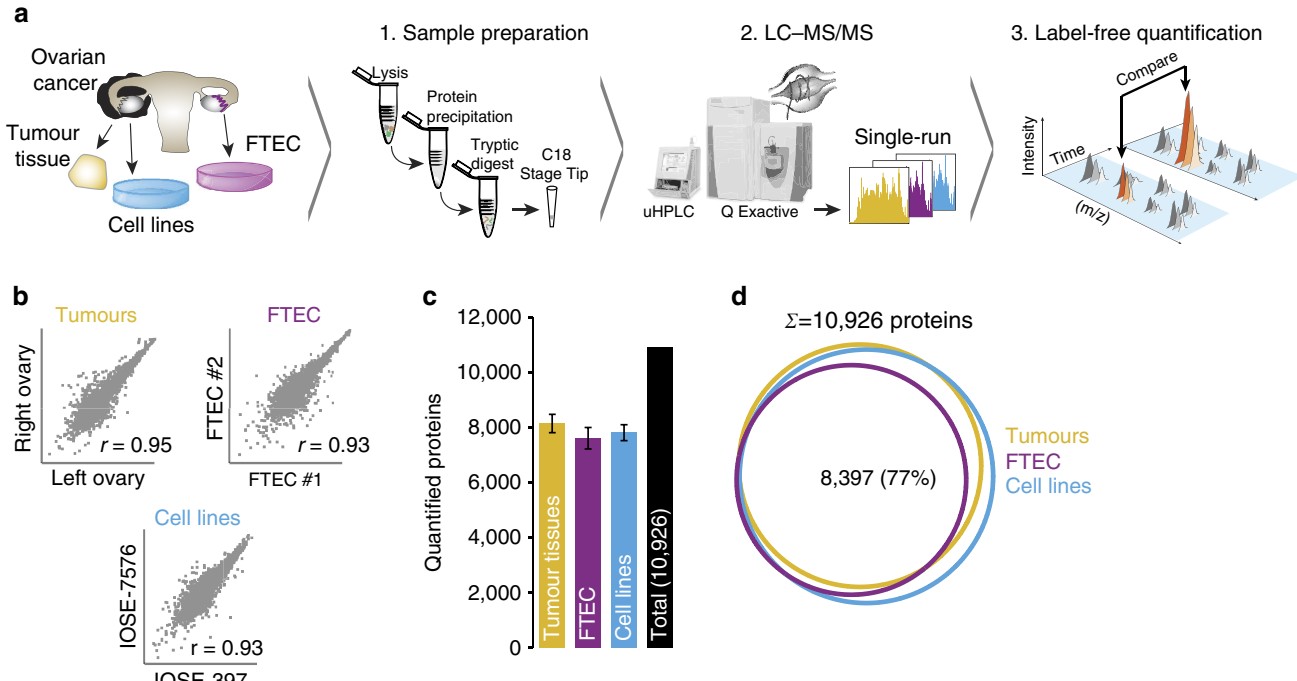

**Figure 1 | Deep single-run proteomics of cell lines and human ovarian cancer tissue.** (**a**) Summary of the shotgun proteomics workflow for OvCa cellular models and high-grade serous ovarian cancer (HGSOC) tissues. Following lysis, protein purification, and tryptic digest, peptides were separated by ultra-high performance liquid chromatography and measured in single runs using a quadrupole Orbitrap mass spectrometer. Label-free proteome quantification was performed using the MaxQuant software environment. (**b**) Workflow reproducibility for cell lines ($n=30$), HGSOC tissues ($n=8$) and primary FTEC cells ($n=3$). Pearson correlations ($r$) were calculated for biological replicates of cell lines, primary FTEC isolates from different healthy donors, and ovarian tumour tissues from both ovaries from a woman with HGSOC. (**c**) Average number of quantified proteins from each sample type. Error bars represent standard deviations. (**d**) Number of proteins common to FTECs, HGSOC tumours and cell lines. FTEC, fallopian tube epithelial cell; LC-MS/MS, liquid chromatography tandem mass spectrometry; *m/z*, mass-to-charge ratio; uHPLC, ultra-high performance liquid chromatography.

across 30 cell lines. Average filtered quantification depth and expression levels from the triplicate analyses of each cell line are shown in Fig. 2a and Supplementary Data 1. We asked which proteins and pathways were constant or differentially regulated across the cell lines and, to that end, calculated the protein level variability (Sd of logarithmized protein intensity) and plotted it against the estimated absolute protein abundance (sum of the intensity-based absolute quantification [iBAQ][20] values calculated by MaxQuant, Fig. 2b). Proteins involved in household functions, such as ribosome biogenesis, were stably expressed across cell lines (median Sd < 0.18), whereas proteins known to be frequently lost, amplified or over-expressed in diverse OvCa subtypes[7,9,21] showed strong expression differences (shown in blue in Fig. 2b). To investigate this in more detail, we focused on known cell line-specific genomic alterations, such as the amplification of the receptor tyrosine kinase *ERBB2* in SKOV3IP1 or the proto-oncogene *KRAS* in KURAMOCHI[9]. Indeed, our data clearly reflected the amplification of these two genes in the expression profiles of those cell lines. Our data also indicated higher expression of KRAS in the carboplatin-resistant TOV112DR, compared with its parental cell line (Fig. 2c). Of note, the proteomic approach can detect protein upregulation regardless of the mechanism responsible. For instance, the transcription factor hepatocyte nuclear factor 1-beta (HNF1B), often over-expressed in clear cell ovarian cancer (CCC)[22] showed the strongest expression in the CCC OVISE cell line and the IGROV1 cell line, which, according to its genomic profile, may be of clear cell or endometrioid origin[9]. Furthermore, as shown previously, the proteomic data allow investigation of the expression status of all proteins in the amplicon[23].

The complete data set covered a large proportion of the KEGG-annotated members of major biological processes and cancer-related signalling pathways, such as all 35 members of the DNA replication pathway and >70% of the p53 pathway (Fig. 2d). Pathway enrichment analysis across the cell lines identified the biological pathways that were most differentially regulated (Fig. 2e). For the quantified proteins that varied the most (Sd > 0.5, 13% of all proteins; above dashed line in Fig. 2b), significantly enriched annotations revealed major differences in the expression of proteins related to the categories 'Extracellular', 'Metabolism', 'Immunity' and 'Adhesion', suggesting pronounced cell line heterogeneity for a variety of biological processes (Fig. 2e, Supplementary Data 2).

**A discriminating 67-protein cell line signature.** Unsupervised hierarchical clustering based on the expression of 8,487 distinct proteins quantified in at least 10 of 30 cell lines resulted in three main groups (Fig. 3a). Group I cell lines comprised OVKATE, SNU119, JHOS4, OVCAR3, COV318, OVSAHO, KURAMOCHI, CAOV3, OVCA433, JHOS2 and the cell line pair PEO1/4, which was derived from primary and recurrent tumours of the same patient and clustered together in the dendrogram. Much of group I consisted of cell lines that were previously reported to likely represent HGSOC cell lines based on features of their genomic profiles such as *TP53* mutational status, mutation frequency and DNA copy number alterations[9]. Group II contained OVCAR5, the cervical cancer cell line pair ME180/C13, the CCC cell line OVISE and the IGROV1 and SKOV3IP1 cell lines; the latter two lines have been previously described as hypermutated and 'unlikely HGSOC' cell lines[9]. In group III, the two IOSE cell

lines, IOSE-397 and IOSE-7576, clustered closely together suggesting that proteomic profiles accurately reflected tissue of origin and/or subtype-specific molecular signatures (Fig. 3a). The IOSE cell lines and the OvCa cell lines HEYA8, DOV13 and 59M had very high proteome similarities (Pearson correlation 0.85–0.92) and they grouped together in a sub-cluster of group III. This group also contained the TOV112D and A2780 cell line pairs, previously described as endometrioid in origin[9,24,25]. In each of their respective groups, the corresponding members of the isogenic cell line pairs were all located directly next to each other

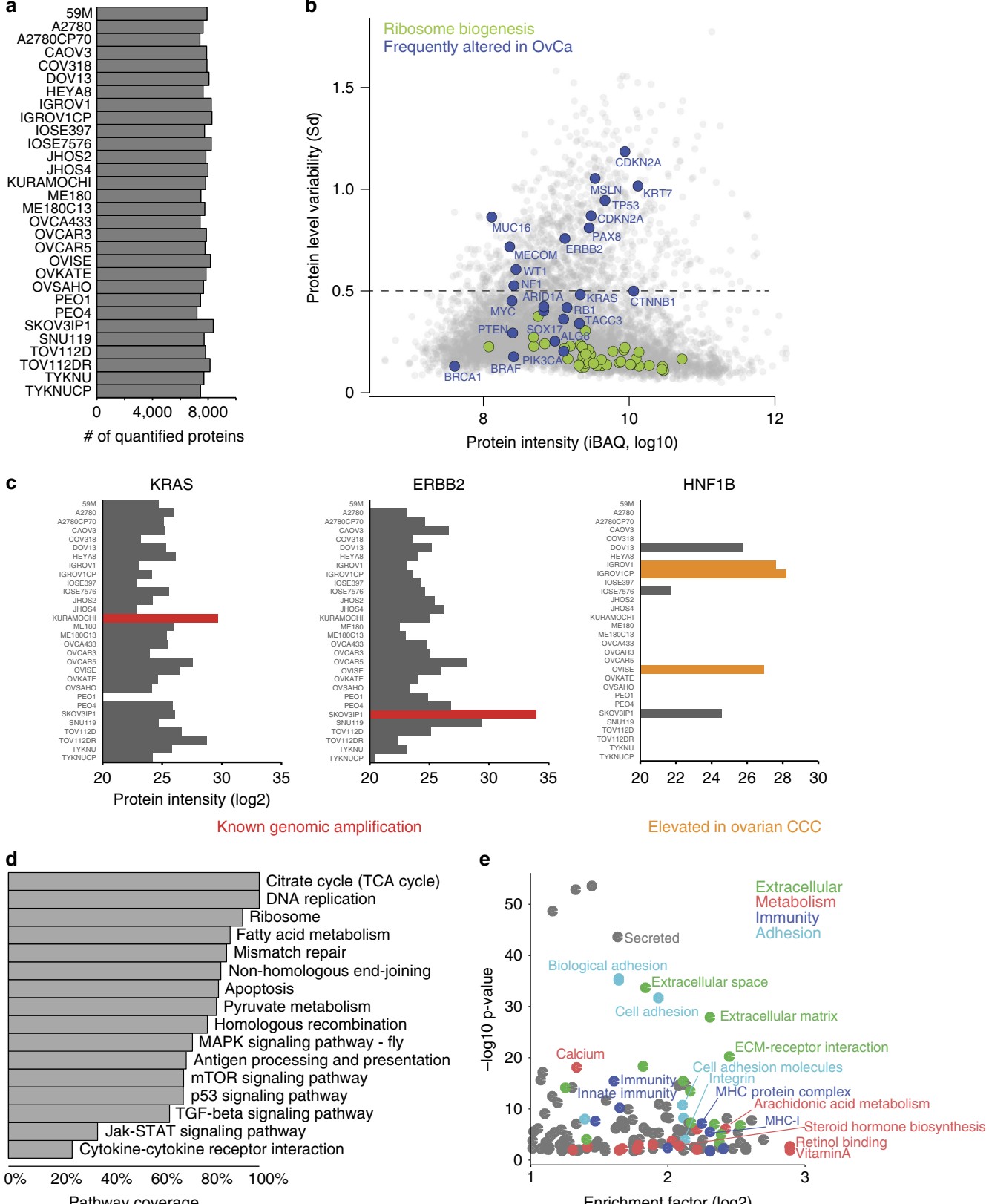

in the clustering analysis. Noticeably absent from group I, and instead clustered within group III, based on their proteomic signature, were the TYKNU and 59M cell lines, both of which display high genomic similarity to HGSOC tumours[9]. The clustering of these two cell lines within group III indicated that certain discriminating features were detectable only at the protein level, and suggested the presence of two distinct HGSOC proteomes.

Independent of the unsupervised hierarchical clustering, we used a principal component analysis (PCA) on the basis of whole-proteome levels, which confirmed the presence of the three main cell line groups (Fig. 3b). Figure 3c depicts the proteins driving the segregation into the three groups: group I proteins included known HGSOC markers such as PAX8, MSLN, KRT7 and MUC16 (CA-125); group II, which contained the OVISE CCC cell line, showed the highest expression of AKR1C1 and HNF1B, known CCC-associated proteins[22,26]; while group III drivers included the mesenchymal proteins HMOX1, VIMENTIN, FN1 and ITGA5 (ref. 7), a protein detected in ~20% of serous OvCa tumours[27].

With the aim of determining a small group of proteins with the strongest discriminating power between the groups, we used feature selection in combination with Support Vector Machines (SVMs) classification, as previously described[28,29]. This identified a set of 67 proteins, which included many known OvCa markers as well as novel ones (Supplementary Fig. 2, Supplementary Table 1).

**Integrative analysis of HGSOC tissue and cell line proteomes**. Given the distinct proteome characteristics of the cell line groups, we reasoned that group I cell lines would most closely resemble HGSOC human cancer tissue. To first assess the quantitative robustness of our workflow for tumour tissue analysis, we analysed eight HGSOC tumour proteomes from five patients. This revealed that the proteomes were very similar between bilateral ovarian tumours from the same patient (mean Pearson correlation 0.95), compared with the inter-patient variation (Pearson correlation 0.72–0.87; Supplementary Fig. 3a). Consistently, tumours from the same patient tightly grouped together in the PCA (Fig. 4a). Adding the cell line proteomes to the PCA indicated a slight gravitation of the group I cell lines towards the HGSOC tumours and the FTECs in component 1 (Fig. 4b), suggesting underlying proteomic similarities with both the OvCa tumours and the primary FTECs. Of note, group III cell lines, which included the two IOSE cell lines, clustered furthest away from both the HGSOC and FTEC proteomes on component 1, but closest to the HGSOC tumours on component 2. Further analysis of component 2 revealed that it represented epithelial/ mesenchymal protein levels (Supplementary Fig. 3b). Application of the discriminating 67-protein cell line signature to the PCA

and hierarchical clustering analyses then resulted in three groups and two main clusters (Fig. 4c,d, Supplementary Fig. 3c). The first cluster was exclusively composed of the group I cell lines, with seven of the eight HGSOC tumours and the primary FTEC samples. In particular, the cell lines COV318, KURAMOCHI and OVSAHO clustered closest to the tumour samples, whereas the PEO1 cell line clustered with the FTEC isolates. The second cluster contained groups II and III cell lines, organized into distinct sub-clusters, and HGSOC-5, which clustered closest to the group III cell lines.

Known epithelial HGSOC markers such as MSLN, PAX8 and KRT7 were higher in the FTEC isolates, HGSOC tumours and group I cell lines compared with groups II and III cell lines (including IOSEs; Fig. 4d). Other proteins with a similar trend in expression included proteins such as the retinoic acid transporter CRABP2, a protein important in cancer-relevant arginine metabolism, ASS1 (ref. 30), and the p53 target gene CRYAB[31]. Conversely, groups II and III cell lines had higher expression of ITGA5, a protein detected in ~20% of serous OvCa tumours[27]. Interestingly, while HGSOC-5 expressed similar levels of the above-mentioned epithelial proteins, it had higher levels of ITGA5, indicating that ITGA5 contributed to its cluster location with group III cell lines.

Cluster-specific protein levels were validated for selected proteins: cluster 1, CRABP2 and PAX8; cluster 2, ITGA5. We confirmed cell type-specific expression of CRABP2 in human HGSOC tumours and normal OSE using immunofluorescence (Fig. 4e), as well as in normal FTECs, fallopian tube HGSOC and omentum HGSOC (Supplementary Fig. 4a). The absence of CRABP2 staining in normal OSE was consistent with its low protein levels in the proteomic profiles of the IOSE cell lines. Furthermore, we confirmed the levels of PAX8, CRABP2 and ITGA5 by western blot (Supplementary Fig. 4b).

**The signature separates HGSOC from TCGA into two groups**. The above analyses established two very robust, distinct clusters based on the 67-protein signature, with cluster 1 containing the group I cell lines, which had an FTEC-type profile and harboured mostly epithelial proteins and cluster 2 containing the group III cell lines, which had an IOSE-type profile with mostly mesenchymal proteins. To validate our finding in a larger patient cohort, we made use of the publicly available data from the Clinical Proteomic Tumor Analysis Consortium (CPTAC)[32,33]. The PNNL proteomics data set was downloaded and we applied our 67-protein signature to the 84 tumour proteomes alone, and then in conjunction with our own proteome data. This confirmed the presence of two main tumour clusters, which we denoted TCGA-A and TCGA-B, and which were enriched for group I/epithelial and group III/mesenchymal proteins, respectively (Fig. 5a, Supplementary Fig. 5). The group I cell

**Figure 2 | Proteomic analysis reveals proteome diversity across frequently used OvCa cell lines.** (**a**) Total number of quantified proteins in 26 OvCa cell lines, two immortalized ovarian surface epithelial cell lines (IOSE) and two cervical cancer cell lines ($n = 3$ measurements for each cell line). A median depth of 7,812 quantified proteins across all the samples was obtained. (**b**) Quantification of proteins with low and high standard deviation (s.d.) across all the 30 cell lines identified constantly or variably expressed proteins. Protein level variability (s.d., log10) and protein intensity (sum of the intensity-based absolute quantification (iBAQ) values, log10, calculated by MaxQuant) were compared for each protein in all cell lines. An s.d. cut-off of 0.5 was used to identify proteins with the highest variability (13% of total proteins; s.d. > 0.5; dashed line). Proteins frequently altered in OvCa are highlighted in blue. Proteins associated with ribosome biogenesis (Gene Ontology Biological Process), shown in green, display small differences in expression across cell lines. CDKN2A is depicted twice, representative of two different isoforms. (**c**) Known genomic alterations are captured at the protein level. The relative protein intensities (MaxLFQ) for proteins whose genes have known amplification events in OvCa cell lines are depicted in red. HNF1B, an ovarian CCC marker, is shown in orange. (**d**) Coverage of cancer-related KEGG pathways. KEGG annotations, including cancer-related pathways, were applied to the data. Percentages of pathway coverage in the indicated cancer-related KEGG categories are shown. Pathways involved in DNA replication and DNA repair, as well as metabolic annotations such as the citric acid cycle or fatty acid metabolism were almost completely covered (>80%). Signalling pathways such as mTOR, p53 or TGF-β signalling are largely covered by >60%. (**e**) Pathway enrichment analysis using Fisher's exact test (Benjamini–Hochberg false discovery rate <2%) was performed for the proteins with the highest expression variability (proteins above dashed line in **b**).

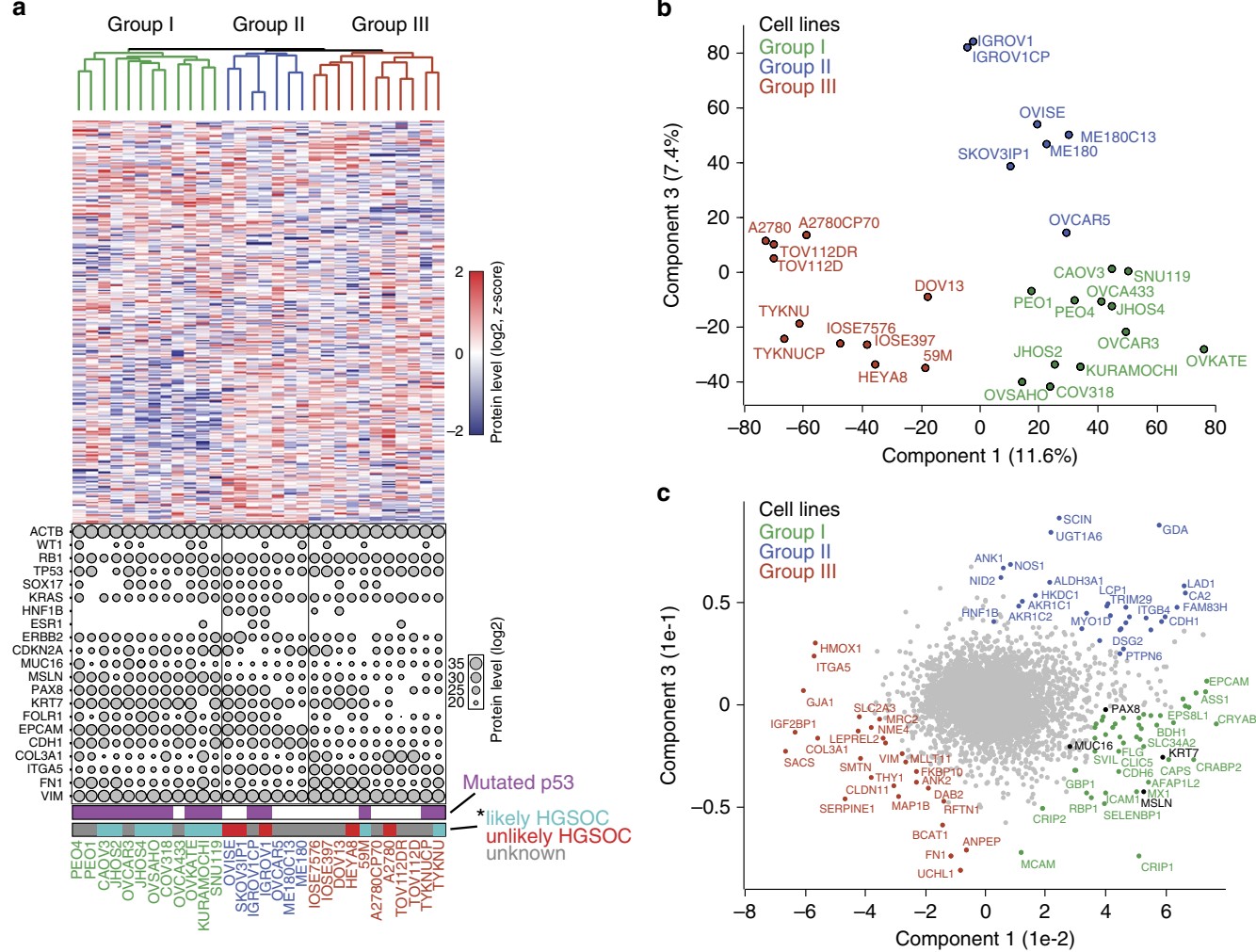

**Figure 3 | Proteomic clustering reveals three distinct cell line groups. (a)** Unsupervised hierarchical clustering was carried out on the basis of the relative expression of 8,487 proteins, delineating three major cell line groups (group I, green; group II, blue; group III, red). The relative expression levels of 15 frequently altered proteins in OvCa are plotted for each of the 30 cell lines. Levels of known OvCa-related proteins (VIM, FN1, EPCAM, CDH1 and CDKN2A) are also plotted. Relative protein levels are depicted by circle size. Relative ACTB levels are included for reference. * 'likely HGSOC' (teal) and 'unlikely HGSOC' (red) refer to the descriptions of these cell lines based on their genomic profiles as reported by Domcke et al.[9]; 'unknown' cell lines in grey were not analysed in that study. **(b)** The presence of three major cell line groups was confirmed by principal component analysis (PCA) of the 30 cell lines. **(c)** Proteins driving the PCA separation. The driver proteins from the three groups are identified on the PCA. Known HGSOC proteins MUC16, PAX8, KRT7 and MSLN are highlighted in black.

lines and the FTECs clustered with the TCGA-A tumours, and the group III cell lines and the IOSEs clustered with the TCGA-B tumours. TCGA-A tumours showed higher expression of most group I/FTEC epithelial proteins such as KRT7, MSLN, CDH6, ASS1 and EPCAM, while TCGA-B tumours had higher expression of the group III/IOSE mesenchymal proteins ITGA5, HMOX1, SMTN and GJA1 (refs 7,34; Fig. 5b). Kaplan–Meier analysis of the overall survival of the two TCGA groups revealed that patients in the TCGA-B group had a significantly lower overall survival than those in the TCGA-A group ($P = 0.0048$; Fig. 5c). We then utilized a publically available messenger RNA (mRNA) data set that had identified two different primary ovarian cancer cell line clusters. In that study, the gene expression profiles of these clusters correlated with different survival outcome in patients[35]. Figure 5d shows that mRNA levels of our group I and group III proteins were higher in the good survival- and poor survival-associated sets, respectively. Thus, the two cell-type clusters established in our integrated cell line, primary cell and tumour proteome analyses are independently validated by the TCGA and Ince[35] data sets.

**Utility of the proteomic resource in functional assay design.** A cell line representative of human HGSOC should not only resemble its tumour of origin at the molecular level, but more importantly, also demonstrate functional utility in pre-clinical investigations. To exploit our global protein expression data, we first determined all proteins that were significantly different between the cluster 1 and cluster 2 cell lines identified in our study and then bioinformatically determined annotation terms that were statistically enriched in either cluster[36] (Fig. 6a,b). Interestingly, this revealed that the vitamin A and retinal binding pathways were the two most enriched pathways in the group I cell lines compared with groups II and III cell lines. In contrast, groups II and III cell lines expressed higher levels of proteins associated with cell proliferation (mitosis, DNA replication, Fig. 6a), in line with a previous study[25]. The volcano plot demonstrates the protein expression fold change between the two clusters. As expected, members of the 67-protein signature showed strong expression differences (Fig. 6b), and the levels of the retinoic acid pathway proteins differed by up to 30-fold. Mean expression of the five vitamin A

pathway proteins likewise showed a large difference between group I and groups II and III cell lines, however, no pronounced differences were present between FTECs, HGSOC tissues and group I cell lines (Supplementary Fig. 6a).

On the basis of the results of this pathway analysis, we hypothesized that group I cell lines would be more susceptible to the known CRABP2-mediated anti-oncogenic effect of all-*trans* retinoic acid (ATRA)[37], the predominant physiological form of retinoic acid, than groups II and III cell lines. This was supported

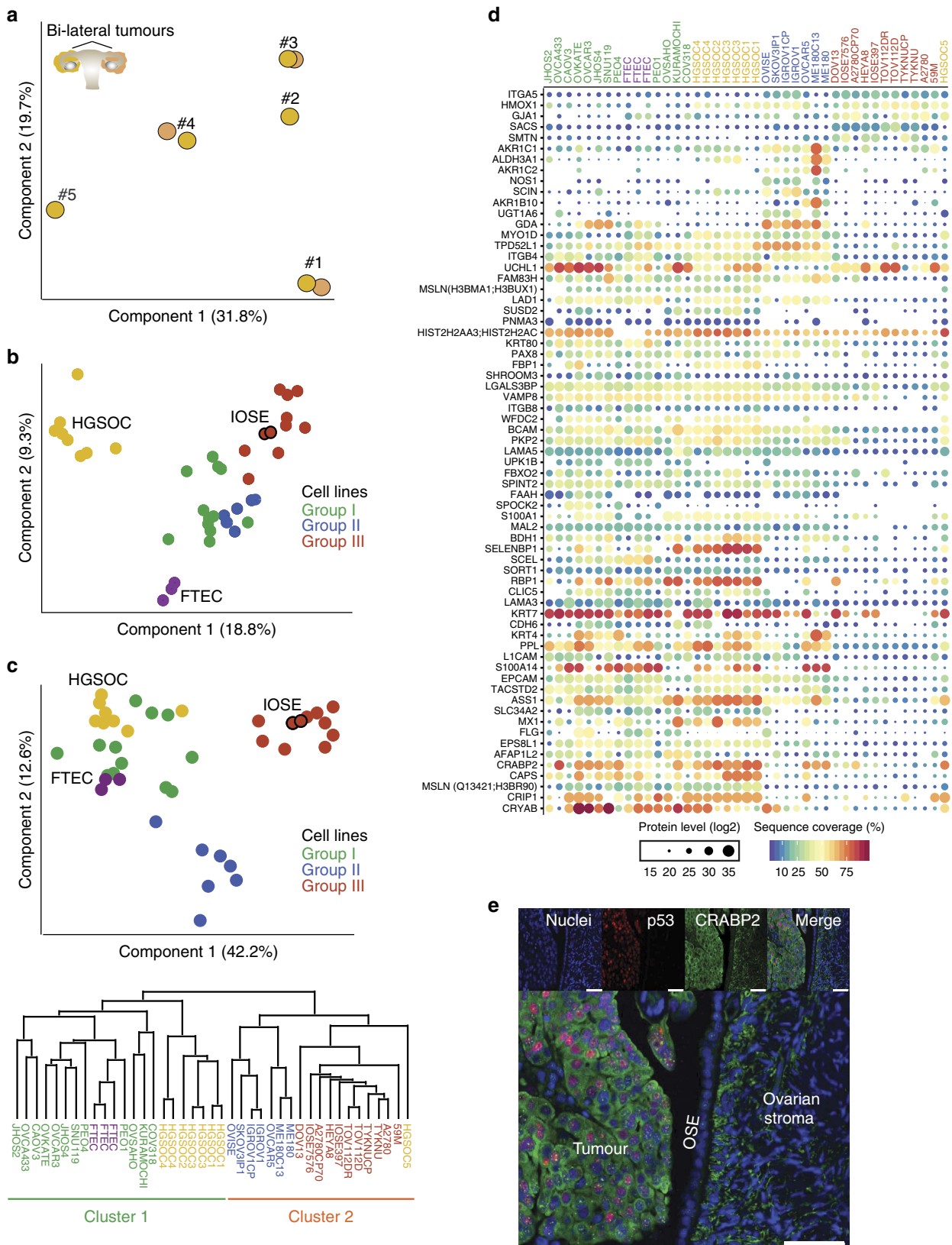

by the inverse correlation between CRABP2 and FABP5, an intracellular lipid binding protein that binds to retinoic acid in the presence of a low CRABP2/FABP5 ratio[38]. The majority of groups II and III cell lines had a low CRABP2/FABP5 ratio (Fig. 6c). To test the hypothesized differential response to ATRA, we treated representative groups I, II and III cell lines with ATRA for 7 days and measured proliferation. Cell proliferation of group I cell lines was inhibited by 26 ± 2% revealing that ATRA induced an anti-oncogenic effect in these cell lines (Fig. 6d). The only exception was the group I KURAMOCHI cell line; however, in these cells, ATRA treatment induced a more differentiated phenotype with long spindle-like protrusions (Supplementary Fig. 6b) consistent with the ATRA-induced differentiation previously reported in other cancers[39,40]. As predicted, ATRA had no inhibitory effect on cell proliferation in the tested groups II and III cell lines; in fact, ATRA-treated HEYA8 and 59M cells showed an increase in proliferation compared with their non-treated counterparts (Fig. 6d).

## Discussion

It is evident by now that the most commonly used HGSOC cell lines are among those with the least genomic similarity to HGSOC tumours[9,11,12]. Nevertheless, these 'bad' cell lines still closely recapitulate the metastatic distribution of human HGSOC in pre-clinical animal models[11].

Here we used MS-based proteomics to offer a more phenotype-associated characterisation of OvCa cell lines than that of genomic and gene expression characterization alone. Our streamlined, single-run workflow allowed us to characterize a large number of cancer proteomes in a relatively short time (4 h). Importantly, the absence of fractionation did not unduly compromise depth of coverage of the proteome as we detected ~8,000 proteins in cell lines, primary cells and tumours, with a total number of ~11,000. To put this in perspective, a recent proteomic study of cancer tissues using a different platform consistently detected 2,000 proteins[41] and our own laboratory previously achieved the numbers detected here only after using fractionation techniques, which greatly add to measurement time and sample requirements[29]. On the basis of the broad and quantitative coverage, the 30 cell lines unambiguously separated into three specific cell line groups: group I, containing a number of HGSOC cell lines, expressed higher levels of many known epithelial HGSOC proteins; group II, containing the CCC cell line OVISE, expressed a number of CCC-associated proteins[26,27]; and group III, containing the IOSE cell lines, expressed relatively lower levels of known epithelial HGSOC markers and higher levels of several mesenchymal proteins. A novel, maximally discriminating 67-protein signature faithfully segregated the groups and contained interesting group-enriched proteins in

addition to known OvCa markers. Of note, while group II cell lines were all cultured in DMEM, the culture media for group I and group III cell lines, both of which contain HGSOC cell lines, consisted of a variety of different media, suggesting that the signature is not dependent on the culture media.

We integrated the cell line proteomes with our primary FTEC and HGSOC tumour data and, in a second step, with the CPTAC proteomic data from 84 patients. Application of the 67-protein cell line signature divided the entire data set into two core clusters, clearly placing the primary FTEC isolates in the epithelial cluster and the IOSEs in the other, more mesenchymal, cluster. The FTEC cluster expressed high levels of known HGSOC proteins such as MUC16 (CA-125), PAX8 and MSLN. It also revealed novel markers for FTEC-derived HGSOC cell lines such as CRABP2 and ASS1, which are highly expressed in serous OvCa compared with the clear cell, endometrioid and mucinous subtypes[30,42]; CRYAB, a p53 target gene[31], which is associated with patient outcome in serous, but not non-serous, OvCa[43]; and CAPS and MX1, which have not been previously described as potential markers of HGSOC (see summary in Supplementary Table 1). The defining feature of the IOSE cluster was the high expression of a small set of mesenchymal proteins: GJA1, HMOX1, ITGA5, SMTN and SACS[7,34]. GJA1 facilitates cell adhesion, invasion and metastasis in a number of other cancers[44–46]; its high expression in the IOSE cluster indicates that it may play a similar role in these cell lines and tumours. $\alpha_5$-integrin (ITGA5), which we previously reported to be an important mediator of early OvCa metastasis[47], was a strong discriminator between the FTEC and IOSE clusters, with high expression in the latter. $\alpha_5$-integrin is regulated by the epithelial differentiation marker E-cadherin (CDH1; ref. 48) and the absence of CDH1 is a predictor of poor survival in OvCa patients[49]. Although CDH1 was not in the discriminating 67-protein signature, its expression was drastically lower in group III cell lines compared to group I cell lines (Fig. 3a). In addition, OSE lacks, or inconsistently expresses, CDH1 (ref. 50). This suggests that the ITGA5/CDH1 axis may be an important distinguishing feature of our newly defined HGSOC sub-groups.

Group II cell lines were the least HGSOC-like cell lines in this study and, interestingly, the ITGA5/CDH1 axis does not appear to apply to this group. In general, group II lines express high levels of epithelial proteins such as CDH1 and EPCAM, which contributes to their clustering with the group I (epithelial) cell lines in component 1 of the cell line PCA (Fig. 3b,c). However, they also express high levels of ITGA5, which may contribute to their clustering with group III in the integrated tumour and cell line analysis (Fig. 4d). At least four of the seven cell lines in group II contain an ARID1A mutation[9]; previous studies have reported an association between these mutations and the transformation of

**Figure 4 | Integrative analysis of HGSOC tissue and cell line proteomes. (a)** Proteomic clustering of eight HGSOC tissue specimens. PCA was performed on the ovarian tumour specimens ($n = 8$) based on the proteomic profiles to evaluate inter-patient heterogeneity and intra-patient homogeneity. Component 1 and component 2 account for 51.5% of the total data variation. **(b)** PCA clustering of cell lines ($n = 30$), FTEC ($n = 3$) and HGSOC tissue proteomes ($n = 8$). Groups I (green), II (blue) and III (red) are as defined in Fig. 3. The juxtaposition of the IOSE cell lines, red circles outlined in black, relative to the rest of the group III cell lines is indicated. Component 1 and component 2 account for 28.1% of the total data variation. **(c)** PCA segregation and clustering of all samples based on the 67-protein signature. The juxtaposition of the IOSE cell lines, red circles outlined in black, relative to the rest of the group III cell lines is indicated. Component 1 and component 2 account for 54.8% of the total data variation. The dendrogram below the PCA summarizes the hierarchical clustering analysis of all samples based on the 67-protein signature. Two main clusters were obtained based on this signature (detailed in Supplementary Fig. 3c). **(d)** Relative levels of the 67 proteins used for PCA clustering in **c**. Relative protein levels (MaxLFQ intensities, log2) are depicted by circle size. Colours indicate protein sequence coverage per sample. MSLN is depicted twice, representative of two different isoforms (shown in parentheses). **(e)** Immunofluorescence staining for CRABP2 and p53 in formalin-fixed paraffin-embedded (FFPE) sections of normal OSE and ovary HGSOC tumour. FFPE sections were stained with an anti-CRABP2 antibody and an anti-p53 antibody, and detected with Alexa Fluor 488- and Alexa Fluor 647-labelled antibodies, respectively. Merged images on the bottom show invasive HGSOC, normal OSE and normal ovarian stroma in the same frame. Scale bar, 50 μM.

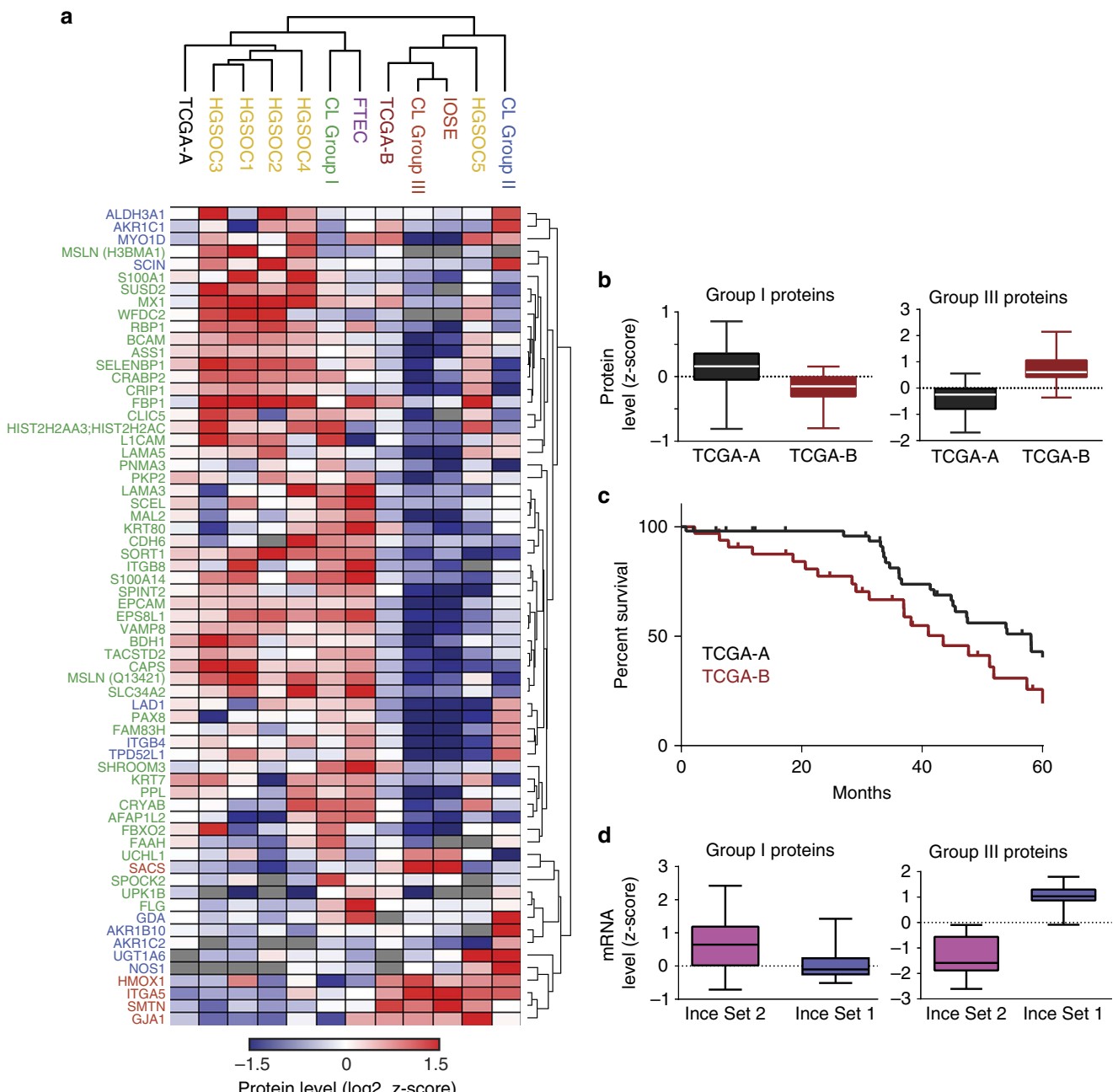

**Figure 5 | Clustering of the TCGA tumours based on the 67-protein cell line signature.** Hierarchical clustering summary of 84 TCGA tumours with the cell lines, FTECs, IOSEs and HGSOC-1 to -5. (**a**) Hierarchical clustering based on the 67-protein signature was applied to the publically available proteomic profiles of 84 TCGA tumours, and our own data set comprises the cell lines, FTECs, IOSEs and HGSOC-1 to -5 (Supplementary Fig. 5). The average protein levels for each sample group are shown. (**b**) Group I and group III proteins are higher in TCGA-A and TCGA-B tumours, respectively. Z-scored protein levels of group I and group III proteins were plotted as box plots for TCGA-A and TCGA-B tumours. (**c**) TCGA-B tumours (maroon line) are associated with a poorer overall survival compared with TCGA-A tumours (black line). Kaplan–Meier overall survival curves were plotted for patients in TCGA-A or TCGA-B. Survival associated with TCGA-B was significantly lower than that associated with TCGA-A (43.5 months versus 58 months, Mantel–Cox $P$ value $= 0.0048$). (**d**) The mRNA levels of group I and group III proteins were analysed in a publically available mRNA data set containing two different primary ovarian cancer cell line clusters[35]. Box plots show higher median mRNA levels of group I proteins in the favourable (Ince Set 2) and relatively lower in the unfavourable (Ince Set 1) survival-associated sets, while the converse is true for group III proteins.

endometriosis into ovarian CCC[51]. In addition, the high levels of CCC proteins, AKR1C1 and HNF1B[22,26], that are low or undetected in the other cell line groups, is further evidence of a potential CCC background for most of group II cell lines.

A 219 microRNA (miRNA)-associated mesenchymal gene signature separated OvCa cases in different data sets into two subtypes: an integrated epithelial subtype and an integrated mesenchymal subtype[34]. In line with this, the proteomes of group I HGSOC cell lines, which clustered with the FTECs, contained a mix of epithelial and mesenchymal proteins, while those of group III, which contained the IOSE cell lines, were predominantly mesenchymal (high VIM, low CDH1 and EPCAM, Fig. 3a). This is also consistent with data from a study showing that the gene expression profile of the mesenchymal HGSOC subtype is similar

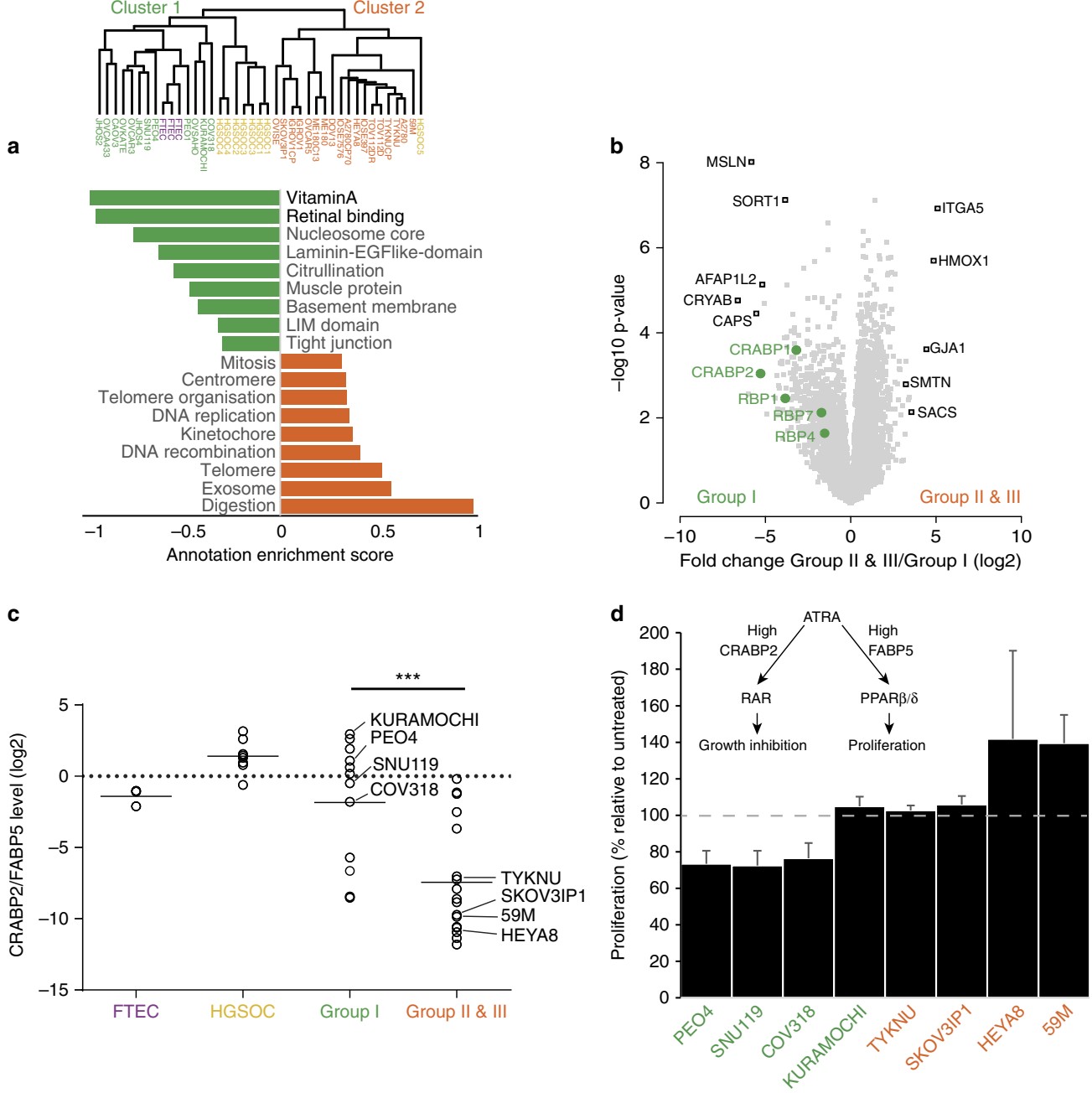

**Figure 6 | Proteomic profiles predict functional cell line properties.** (**a**) Pairwise comparison of enriched annotations for group I (cluster 1) and groups II and III (cluster 2) cell lines. Pathway enrichment analysis was calculated on the basis of the protein expression fold change between group I and groups II and III cell lines. Green and orange bars denote the strongest enriched pathways (Benjamini–Hochberg FDR < 0.02) in group I and groups II and III cell lines, respectively. Annotation enrichment position score, between −1 and 1, indicates the centre of the protein distribution for each significant category, relative to the overall distribution of values. Dendrogram shows sample clustering, based on the 67-protein signature. (**b**) Volcano plot of the pairwise comparison between group I and groups II and III cell line proteomes. Expression fold changes (*t*-test difference, log2) were calculated and plotted against the *t*-test *P* value (−log10). Vitamin A pathway-associated proteins are highlighted in green. Their position on the left side of the plot indicates their higher expression in group I cell lines (*P* = 0.003). Strongest outlier proteins for both groups are marked in black. (**c**) Relative protein levels of the retinoic acid transporter proteins CRABP2 and FABP5 were compared between FTECs, HGSOC, group I cell lines and groups II and III cell lines. The CRABP2/FABP5 ratios are shown. Groups II and III cell lines show significantly lower levels of CRABP2 expression relative to FABP5 (two-sided *t*-test, *P* value < 0.001). The cell lines used in subsequent proliferation experiments (d) are indicated. (**d**) ATRA treatment induces growth arrest or differentiation in group I cell lines. Group I (green) and groups II and III (orange) cell lines were treated with 7 μM ATRA daily for 7 days. On day 8, MTT assays were performed. Results are plotted as the proliferation rate of ATRA-treated cells relative to that of untreated cells. Values < 100% indicate growth arrest; values > 100% indicate growth promotion. Error bars represent standard errors for two to four replicates per cell line. A simplistic model of the ATRA pathway is shown. ATRA, all-*trans* retinoic acid; PPAR, peroxisome proliferator-activated receptor; RAR, retinoic acid receptor.

to that of normal ovarian tissue[52]. Our separation of HGSOC tumours into two groups on the basis of the 67-protein signature was solidly confirmed in a validation set from the CPTAC as it grouped their 84 HGSOC tumours into two main clusters, reflective of group I and group III cell line proteomes, also providing further evidence for a dualistic precursor model of HGSOC. Importantly, there is a clear survival difference between the two TCGA groups, with the TCGA-B (mesenchymal) patients demonstrating a significantly worse overall survival than that of the TCGA-A (epithelial) patients. In addition, the juxtaposition of our tumour specimen HGSOC-5 with group III cell lines, the IOSEs and TCGA-B in the integrated analysis strongly indicates that this tumour represents the mesenchymal HGSOC subtype[7,52].

Clustering of group I cell lines and HGSOC-1 to -4 tumours with the FTECs provides support for the fallopian tube epithelium as their cell of origin, while the clustering of group III cell lines and HGSOC-5 with the IOSEs, and their lower expression of epithelial proteins, suggests that they may be OSE-derived. This suggests the potential presence of an inherent level of stratification in HGSOC tumours based on their protein expression and cell of origin. Regarding the precursor cell of HGSOC, there are convincing arguments for both FTECs and OSE[3,4,53,54]. Supporters of the 'fallopian tube theory' argue that the HGSOC protein PAX8 is a marker of the tubal epithelium[55] and that STICs in the fallopian tube fimbria of BRCA1-mutation carriers[4] and HGSOC patients[3] represent part of the serous carcinogenic sequence; supporters of the 'ovarian theory' argue that OSE expresses multiple stem cell markers[56], and that OSE-lined inclusion cysts can undergo tubal metaplasia[56,57] followed by transition to carcinoma[58]. There is support for both hypotheses from animal models[54,59]. However, aside from the investigation of STICs, distinct biologic, clinical or molecular features capable of categorical differentiation between the two have not yet been identified[60]. Our proteomics results now suggest a potential new level of simple stratification in HGSOC into mesenchymal and epithelial subtype HGSOC. However, this is, as yet, only hypothesis-generating; more detailed molecular, immuno-histochemical and clinico-pathologic studies will be necessary to substantiate or reject this concept.

The underlying differences between the cell lines in the two different clusters were also reflected in the biological pathways associated with their respective proteins. The vitamin A and retinol pathways were highly enriched in group I cell lines, with more than 10-fold expression level differences. Retinoic acid has been used successfully in the treatment of acute promyelocytic leukemia and neuroblastoma and its limited success in other cancers may be due to different cellular responses. On the basis of the proteomic profiling alone, we correctly predicted that group I cell lines would be sensitive to the anti-proliferative, differentiating effects of ATRA, while groups II and III cell lines would be either ATRA-resistant, or responsive to its pro-oncogenic effects. Future investigation into these different mechanisms of action of, and cellular response to, ATRA in group I and group III HGSOC cell lines may inform the investigation of retinoic acid in OvCa.

In summary, the integrated level of cell line proteomic profiling introduced here provides the OvCa research community with an additional resource to select the most appropriate model for their research. The described 67-protein signature may shed more light on the cell of origin and respective driver proteins of HGSOC and contribute to the further investigation of important clinical problems such as chemotherapy resistance. Apart from opening the door for systematic and routine interrogation of proteome-wide differences in cancer models, our streamlined and high-sensitivity proteomics workflow will be especially attractive in *in vivo* contexts, where only small numbers of cells are available.

## Methods

**Patient samples.** All patient samples were collected at the University of Chicago Medical Center, with approval from the Institutional Review Board. All the patients provided informed consent.

**Frozen tumour samples.** Ovarian tumours from five chemo-naive patients with serous-papillary high-grade, FIGO Stage IIIB/C, OvCa were collected by EL under an IRB-approved protocol at the University of Chicago during the primary debulking surgery or laparoscopy before neo-adjuvant treatment. In some cases, tumours were collected from both ovaries ($n = 3$ patients). Tumours were immediately snap frozen and stored at $-80\,^\circ C$ until sample processing for MS analysis. Serous histology was confirmed by two gynaecologic pathologists (A.M., R.R.L.).

**Primary FTEC isolation.** Fallopian tubes were removed from three patients with benign gynaecological conditions not affecting the fallopian tube. Primary fallopian tube secretory epithelial cells (FTECs) were isolated[61] and cultured[62] as previously described.

**Cell lines.** Thirty cell lines were included in this study; their sources and respective media are detailed in Supplementary Data 3. All the cell lines were genotyped to confirm their authenticity; cell lines were authenticated by using the commercial service CellCheck (IDEXX Bioresearch). The samples were confirmed to be of human origin and no mammalian inter-species contamination was detected. The alleles for nine short tandem repeat markers were determined and the results were compared with the profiles from DSMZ, ATCC, JCRB and RIKEN short tandem repeat databases. All the cell lines were mycoplasma-negative. The cells were grown under recommended culture conditions and the samples were collected from three consecutive passages for $n = 3$ replicates for each line.

**Sample preparation for MS analysis.** Cell lysis was performed in lysis buffer (4% SDS, 10 mM Hepes pH 8.0) at $99\,^\circ C$ for 10 min and by 15 min sonication (level 5, Bioruptor, Diagenode). HGSOC tissues were first homogenized in lysis buffer using an Ultra Turbax blender. Proteins in the lysate were reduced with 10 mM DTT for 30 min and alkylated with 55 mM iodoacetamide for an additional 30 min. Remaining SDS detergent was removed by acetone precipitation. Briefly, acetone ($-20\,^\circ C$) was added to 100 µg of proteins to a final concentration of 80% v/v and the proteins were precipitated overnight at $-20\,^\circ C$. After centrifugation (15 min, $4\,^\circ C$, 16,000 g), the detergent-containing supernatant was removed and the protein pellet was washed with 80% acetone ($-20\,^\circ C$). Protein pellets were then resolved in 100 µl 6 M urea/2 M thiourea (in 10 mM Hepes pH 8.0) and digested with 1 µg of LysC for 3 h at room temperature. After adding four volumes of 50 mM ammonium bicarbonate, 1 µg trypsin was added and tryptic digestion carried out overnight. The next day, digestion was stopped by adding 1% TFA. Peptides were finally desalted on C18 StageTips and kept at $-20\,^\circ C$ until MS analysis.

**Liquid chromatography-MS analysis.** MS analysis was performed using Quadrupole Orbitrap mass spectrometers[63,64] (Q Exactive and Q Exactive HF, Thermo Fisher Scientific, Rockford, IL, USA) coupled to an EASY-nLC 1000 HPLC system (Thermo Fisher Scientific) via a nano electrospray source. Columns (75 µm inner diameter, 50 cm length) were in-house packed with 1.9 µm $C_{18}$ particles (Dr Maisch GmbH, Germany). Peptides were separated over a 250 min gradient from 2% to 60% (5 min to 5%, 180 min to 25%, 45 min to 35%, 20 min to 60%) in buffer B (80% acetonitrile, 0.5% formic acid) at 200 nl min$^{-1}$. The column temperature was constantly set to $50\,^\circ C$ by using an in-house-made column oven. The survey scans (300 to 1,650 $m/z$) were acquired with a resolution of 70,000 (60,000 for Q Exactive HF), at $m/z$ 200. A top-five method was used to select up to the five most abundant precursor ions with a charge $\geq 2$. Selected precursor ions were subjected to high-energy collisional dissociation fragmentation at a normalized collision energy of 25 (27 for Q Exactive HF), an isolation window of 2.2 Th (1.4 Th for Q Exactive HF) and a resolution of 17,500 at $m/z$ 200 (15,000 for Q Exactive HF). For survey scans, ion injections times were set to 20 ms (target value 3E6) and 120 ms (target value 1E5) for MS/MS scans. Dynamic exclusion of sequenced peptides was set to 30 s. Data were acquired using Xcalibur software (Thermo Scientific).

**MS data analysis.** MS raw files were analysed with MaxQuant software[18] (version 1.5.0.38). MS/MS-based peptide identification was carried out with the Andromeda search engine[17] in MaxQuant. Briefly, Andromeda uses a target-decoy approach to identify peptides and proteins at an FDR <1%. As a forward database, the human UniProtKB database (Oct 2014) was used. A reverse database for the decoy search was generated automatically in MaxQuant. Enzyme specificity was set to 'Trypsin', and a minimum number of seven amino acids were required for peptide identification. Default settings were used for variable and fixed modifications (variable modification, acetylation (N terminus) and methionine oxidation; fixed modification, carbamidomethylation). Proteins and protein isoforms that could not be discriminated by unique peptides were grouped into protein groups[18]. For label-free protein quantification, the MaxLFQ algorithm was used as part of the MaxQuant environment[19]. Briefly, quantitative information was retrieved on the

basis of high-resolution three-dimensional peptide profiles in mass-to-charge, retention time and intensity space. The algorithm first calculated pairwise protein ratios by taking the median of all pairwise peptide ratios per protein. Only shared identical peptides were considered for each pairwise comparison. A minimum number of one ratio count was required for each pairwise comparison. To retrieve quantitative information for all possible sample comparisons, a least-squares analysis was used to reconstruct the relative abundance profile for each protein. This step preserved the total summed intensity for a protein over all the samples. To maximize the number of quantification events across samples, we enabled the 'Match Between Runs' option in MaxQuant, which allowed the quantification of high-resolution MS1 features that were not identified in each single measurement.

Isobaric tag for relative and absolute quantification (ITRAQ)-based TCGA proteome raw files (PNNL study) generated by the Clinical Proteomic Tumor Analysis Consortium (NCI/NIH) were downloaded from the CPTAC data portal (https://cptac-data-portal.georgetown.edu/cptacPublic/) and analysed with MaxQuant. Logarithmic reporter intensities were normalized against the control reporter channel (channel 117, pooled sample of 84 TCGA ovarian tumour tissue samples) and each sample median normalized before data analysis.

**Statistical analysis.** All statistical and bioinformatics analyses were performed using the freely available software Perseus[65] (as part of the MaxQuant environment) or the R framework[66]. Proteins identified only by site modification or found in the decoy reverse database were not considered for data analysis. For the analysis of LFQ cell line data, we first filtered out proteins that were only quantified in one of three replicates and took the average expression per cell line for the remaining protein quantifications. To calculate protein level variability across cell lines (standard deviation of MaxLFQ intensity, Fig. 2b) we required a minimum of 10 quantified values. Pathway enrichment analysis for categorical data (Fig. 2e) was performed based on a Fisher's exact test with a Benjamini–Hochberg FDR threshold of 0.02. GOBP, GOCC, KEGG and Uniprot Keyword annotations were used for enrichment analysis and required a minimum category size of at least four proteins. For numerical data such as protein expression fold change between two groups (Fig. 6b) we used a one-dimensional enrichment analysis[36] with a Benjamini–Hochberg FDR threshold of 0.02. For hierarchical clustering of cell lines (Fig. 3a), a minimum of 10 valid values (one-third of all samples) was required. Missing values were imputed on the basis of a normal distribution (width = 0.15, down-shift = 1.8). MaxLFQ intensities were first z-scored and the samples clustered according to Spearman rank correlations as a distance measure for column and row clustering.

We used the classification framework implemented in Perseus and, in particular, SVMs combined with feature selection methods to identify a subset of proteins that act as strong discriminators between the three cell line groups. This should allow deeper insight into the underlying biological processes, while keeping the selected subset small enough to allow follow-up studies (Supplementary Fig. 2, Fig. 4c,d).

The SVMs implementation in Perseus is an adaptation of the well-established and commonly used LIBSVM library[67], which uses sequential minimal optimization to solve the quadratic problems during model training. We used a One-versus-All implementation of SVMs classification, which resulted in three distinct ranked lists of proteins—one for each of the cell line groups. Feature selection was embedded in a cross-validation procedure to avoid overfitting and random sampling using 85% of the data for training and the other 15% for testing; this was repeated 250 times. Proteins were ranked by the P value computed using a modified test statistic[68] with an $s_0 = 4$ parameter. For the final list of 67 discriminating proteins, the top-ranked ones from each of the three ranked lists that offered a good tradeoff between minimal sets and the smallest error rates were combined (Group I, 53 proteins; Group II, 10 proteins; Group III, 10 proteins; with six proteins overlapping). This feature selection method resembles that of a domain expert (biologist) selecting a small subset of proteins that can conveniently be followed up. Furthermore, the required large differences between the groups should allow more robust classification in a clinical setup. The method is conceptually analogous to the standard analysis of variance test and, in fact, performing analysis of variance identifies the selected proteins as significantly differentially expressed. However, using feature selection avoids the need for an FDR cutoff and allows for selecting smaller subsets.

For PCA of cell lines, primary cells and HGSOC tissues (Fig. 4b), we first filtered out the roughly 200 most abundant plasma proteins[69]. A minimum number of 30 valid values out of 41 was required, resulting in 6,649 proteins. Missing values were imputed as described above.

For the clustering of ITRAQ and label-free data (Fig. 5a,b, Supplementary Fig. 5), z-scoring was performed group-wise for ITRAQ and label-free data.

For pairwise comparison of proteomes (Fig. 6b), a two-sided t-test statistic was used, including a permutation-based FDR of 5% and an $s_0$ value[68] of 2.

**ATRA treatment and proliferation assay.** The following cell lines were treated (or untreated) with 7 μM ATRA daily for 7 days in 96-well plates: SNU119, TYKNU, KURAMOCHI, HEYA8, PEO4, 59M, SKOV3IP1 and COV318. On day 8, MTT assays were carried out. Proliferation rates were calculated for ATRA-treated and untreated cells and plotted as percentage relative to untreated cells.

**Western blot.** The cells were grown in six-well plates and treated with 7 μM ATRA daily for 7 days. The cell lysates were collected in RIPA buffer and 20 μg protein was electrophoresed on 4–20% resolving gels. The following antibodies were used in western blots: anti-rabbit ITGA5 (1/1,000 dilution, #sc-10729, Santa Cruz Biotechnology, Dallas, TX, USA), anti-rabbit PAX8 (1/1,000 dilution, #9857, Cell Signaling Technology, Danvers, MA, USA), anti-rabbit CRABP2 (1/3,000 dilution, #PA5-27451, Thermo Fisher Scientific) and anti-rabbit GAPDH (1/1,000 dilution, #2118, Cell Signaling Technology).

**Immunofluorescence.** Formalin-fixed paraffin-embedded HGSOC and normal tissue sections (5 μm) were deparaffinized in xylene and rehydrated in graded alcohol solutions. One FFPE section of normal (OSE and FTEC) or tumour (omental tumour and fallopian tube tumour) tissue was used for immuno-fluorescence. Antigen retrieval was carried out in 0.01 M sodium citrate buffer with 0.05% Tween 20 (pH 6.0) for 30 min at 95 °C. Briefly, slides were washed 3 × with PBS, blocked with 10% goat serum in phosphate-buffered saline/0.05% Tween (PBST), and incubated overnight with anti-rabbit CRABP2 antibody (1:200 dilution, #PA5-27451, Thermo Fisher Scientific) or anti-mouse p53 (1:200 dilution, pantropic, #OP42, Millipore, Billerica, MA, USA). Following five PBST washes, sections were incubated with Alexa Fluor 488 goat anti-rabbit IgG (1:500 dilution, #A11008, Thermo Fisher Scientific) or Alexa Fluor 647 goat anti-mouse IgG (1:500 dilution, #A21235, Thermo Fisher Scientific) and Hoechst (1/200 dilution, #H1399, Thermo Fisher Scientific) for 1 h, washed 5 × with PBST, 2 × with PBS, and mounted with Prolong Gold Antifade Reagent (P36934, Molecular Probes, Thermo Fisher Scientific). The images were obtained with a Zeiss 510 LSM confocal microscope, using the LSM 510 software. As a negative control, the primary antibody was omitted for one section in each set of samples.

**Data availability.** The mass spectrometry proteomics data have been deposited to the ProteomeXchange Consortium (http://proteomecentral.proteomexchange.org/cgi/GetDataset) via the PRIDE[70] partner repository with the data set identifier PXD003668. The data set is also accessible via the user-friendly MaxQB database (http://maxqb.biochem.mpg.de/mxdb/project/list). All other data supporting the findings of this study are available within the article and its Supplementary Information files or from the corresponding author upon reasonable request.

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

## Acknowledgements

We thank Korbinian Mayr, Igor Paron and Gabriele Sowa for their assistance in mass spectrometric analysis. This work was supported by the Max-Planck Society for the Advancement of Science, the Körber Foundation (Körber European Science Prize), a National Cancer Institute grant (NCI-R01 CA111882—E.L.) and a Colleen's Dream Foundation award (K.M.W.).

## Author contributions

F.C. acquired and interpreted the proteomics data, developed the concept and edited the paper. K.M.W. wrote the manuscript, developed the concept, collected the samples and

carried out the ATRA experiments, western blots and immunofluorescence. M.C. and C.Y.C. collected the samples. M.A.E. isolated the primary cells. S.T. performed the statistical analysis. A.M. and R.R.L. helped with the histological analysis and provided advice. M.M. and E.L. together designed and supervised the study, developed the concept and edited the paper. All the authors have approved the final version.

## Additional information

**Competing financial interests:** The authors declare no competing financial interests.

