## [Peer Review File · Nature Communications]

Editorial note: Reviewer #4 was added in the second round of review to provide specific comments on the support vector machine classification analysis used in this manuscript.

Reviewers' comments:

Reviewer #1 : Ovarian Cancer

(Remarks to the Author):

This is an interesting and well-written report. The investigators appropriately discuss the limitations of cell line research in understanding the biology of human cancers. That being said, the question remains as to the relevance of this research. For example, the discussion of a possible role for ATRA based on this cell line analysis remains questionable. Comment required.

Reviewer #2: Ovarian Cancer

(Remarks to the Author):

In this manuscript, the authors undertake LC-MS/MS based proteomic analysis of 30 cell lines, mainly of ovarian cancer origin, and identify three groups and two clusters of protein expression based upon unsupervised hierarchical clustering and principal component analysis. They also identify a discriminating 67 protein signature.

Group I contains cells lines previously identified as 'likely HGSOC' in the Domcke genomic analysis, group II a mixture of lines including some CCC, some unclassifiable lines and the two Cx lines, whilst group III contains the immortalised IOSE lines, some lines felt unlikely to be HGSOC by Domcke and, crucially, at least two lines (59M and TYCNU) with TP53 mutations felt likely to be HGSOC on genomic analysis. Thus, the data suggest that HGSOC may have two proteomic signatures (exemplified by groups I and III). These data are supported by analysis of 8 primary HGSOC tumour from 5 patients. Analysis of publically available proteomic data from the TCGA sample set also supports a potential binary proteomic division of HGSOC with the 84 samples clustering into two distinct groups (TCGA-A and TCGA-B), with differences in cell of origin (fallopian tube vs ovary) postulated. The authors then suggest that differential expression of retinoic acid components could identify cell lines in groups I and III with differential sensitivity to ATRA treatment.

Overall, the work is of high quality and the MS pipeline is technically impressive. The manuscript is well written and the figures extremely clearly presented. I think that these are important data and will be of interest to the ovarian cancer community. However, there are several key questions raised by the results.

The main question relates to the ultimate utility of the proteomic signatures and how are they to be taken forward. For example, there are several clinical questions - are there prognostic implications of the two different HGSC protein clusters? Is there enough clinical information in the TCGA sample data to allow any prognosis implications to be drawn or even inferred? If it could be shown that the 67 protein signature was prognostic in patient samples, it would increase the significance of the data. Second, can this signature ever be robust enough or simple enough for routine clinical use? If not, is there an obvious genotype/protein phenotype relationship that could be utilised for stratification and/or prognostication?

In addition, Ince et al (Nat. Comms. 2015) undertook gene expression and RPPA analyses of primary cell cultures and some established lines, identifying two major clusters - I cannot see reference to this manuscript here. What overlap exists between the data presented here and those previous data (especially the RPPA analyses)?

Specific points:

1. Figure 2b - 'frequently altered in ovarian cancer' - how was this group of proteins defined?
2. Figure 2c - it is not quite clear what this is trying to show: I agree that there is obvious KRAS expression in KURAMOCHI, but other lines with known KRAS mutations (e.g. HeyA8) don't show up here.
3. Figure 2d - pathway enrichment of the proteins that are most variably expressed: again, what is this trying to demonstrate?
4. In the PCA analysis in Figure 3b, it is not clear why OVCAR5 are grouped in group II not group I.
5. Figure 4a - having 8 tumours from 5 patients strengthens the paper, but 8 is not very many, especially given that one patient (HGSOC-5) appears to behave differently from the others. What is going on with this tumour - is it possible to identify a critical difference between this tumour (e.g. type of p53 mutation, CCNE1 amplification) compared to the others? This also relates to question about the ultimate utility of these data above.
6. I regret that Figure 6b is not terribly convincing.
7. Culture medium. What effect does culture medium have upon protein expression? Ince et al also demonstrated that the nature of culture medium could have profound effects upon growth and even copy number profiles of cells that grew from primary cultures. Have the authors demonstrated the reproducibility of signatures in different culture media?

Minor point

1. Supplementary table 3 is labelled as supplementary table 4.

Reviewer #3: tumour microenvironment and proteomics

(Remarks to the Author):

In this manuscript, Coscia F. and colleagues have used single-run mass spectrometry to perform in-depth proteomic analysis of 26 ovarian cancer cell lines commonly used for experiments, 2 cervical cancer, 2 immortalised ovarian surface epithelial cells, three primary fallopian tube epithelial cell isolates and 7 ovarian high grade ovarian cancer tissues. By using statistics and bioinformatics, the Authors have identified three separated clusters of cell lines with different proteomic signatures and using supporting vector machine they have define a 67-protein cell line signature which separated cell lines into epithelial (containing fallopian tube epithelial cell) and mesenchymal (containing ovarian surface epithelial cells) type, and which was able to similarly segregate patient samples data from CPTAC/TCGA tumour proteome dataset. The Authors hypothesised that their signature can stratify patient samples according to the origin of the ovarian cancer. Moreover, the Authors provide confirmation of CREBP2 levels in the different cell lines using western blot, and CREBP2 expression in high grade serous ovarian cancers using IHC on patient samples. Furthermore, they provide functional validation of retinoic acid pathway regulation using ATRA treatment on ovarian cancer cells.

This is an original work which uses an unbiased and state-of-the-art, highly accurate quantitative approach to answer key open questions in the ovarian cancer fields. The Authors provide first evidence that proteomics has the potential to successfully identify the origin of ovarian cancers. Moreover, they provide the proteomic analysis of almost 30 commonly used ovarian cancer cell lines as valuable resource, which, combined to previously published genomic analysis, will help researchers to select the appropriate cancer cell model for their study. I strongly believe that the robustness of this work and the findings make this manuscript suitable for Nature Communications. However, there are several issues that need to be addressed by the Authors.

Page 5: The Authors claim that for this work they have developed a method based on single-run described recently. What is exactly the development that they have made and how has this improved the previous method?

Page 6 "protein expression values": expression should be replaced by levels, since protein levels do not necessarily depend only on expression, but also degradation and other mechanisms.

Fig S1d: The coverage of pathways related to cancer is interesting information. I would move that panel in the Figure 1 of the manuscript.

Page 7 and Fig 2b: the Authors should explain what iBAQ has been used in the plot. Is that the sum of the iBAQ calculated by MaxQuant for the cell lines?

Fig 2c. To make their statement more solid, the Authors should expand the range of examples and show a broader panel of known deleted, amplified and mutated genes in ovarian cancer cell lines (e.g. based on Ref 9), including for example Myc, Rb1 and p53.

Page 9: proteins mentioned as examples for group II and III seem to have been chosen somehow randomly. Could the Authors rationally explain why they chose them as examples?

Fig 4b and page 10. The Authors comment extensively on component 1. However, component 2 suggests that there are some proteomic similarities between HGSOC and IOSEs in Group III. Please comment on this as well. Considering the Authors' findings later in the manuscript, could component 1 represent cell proliferation?

Page 10. The discussion of ITGA5 and AKR1C1 is not clear. Are the Authors trying to say that the three groups could somehow represent three different ovarian cancer types? Information about ITGA5 and AKR1C1 could have been mentioned already at page 9, and here the Authors could just highlight that those proteins are included in the 67 protein signature.

Fig 4e: S4a and Page 11. The Authors should write somewhere (Methods or Figure legend) how many tissues have been stained to conclude that CRABP2 is not expressed in HGSOC but not normal OSE and FTEC.

Fig S4b: To strengthen their conclusion, in addition to confirming the proteomic data for CRABP2 by western blot, the Authors should provide CRABP2 staining of non-HGS ovarian cancers, such as clear cell and endometrioid ovarian cancers, for comparison to HGSOC.

Page 11: The Authors mention only here for the first time the presence of mesenchymal markers in group III. I would mention it earlier on, when the Authors comment about group I, II and III. Page 12. Provide references for ITGA5, HMOX1, SMTN and GJA1 to be mesenchymal markers.

Page 11-13 and Figs 5 and 6: I find confusing that in the paragraph "Integrative comparison of HGSOC..." the Authors compare Cluster 1 (= group I) with Cluster 2 (=group II and III), then in the following paragraph "HGSOC from TCGA..." they consider group I and group III, and then in paragraph "Utility of the proteomics..." they go back again to compare Cluster 1 with Cluster 2. I think that the comparison between the two clusters (last paragraph) gives a clear rationale on why

to focus on CRABP2 for validation, rather than mentioning (page 11) "we first focused on CRABP2 because retinoic acid pathway was differentially expressed...". Indeed, many other processes were differentially regulated. After the validation of CRABP2, the Authors could do functional validation with ATRA. Then, they could finish with the utility of the 67 protein signature to group patients cancer tissues.

Page 11-13: The relevance/meaning of Group II is not clear. The Author should discuss more clearly what they think this cluster represents. For example, looking at figure 3c, Group II cell lines seem more epithelial-like, since they have high levels of CDH1. However, they have similar levels of ITGA5 compared to group III, which is considered more mesenchymal. The Authors should address this point.

Page 12: The Authors wrote that Cluster 2 contains higher levels of proteins involved in mitosis. Have the Authors checked, or are data already available, the proliferation of these cell lines? Could it be that proliferation is major discriminating factor between Cluster 1 and Cluster 2 rather than the epithelial/mesenchymal status of the cells?

Page 12-13: to strengthen their finding the Authors should perform the ATRA experiment in CRABP2 silenced cells, to show the specificity of the measured proliferation effect. Alternatively they should use a second approach, similar to ATRA.

Page 11. Typo: Supplementary Fig 4a and not 8a.

Data analysis:

Overall, the proteomic analysis (MS samples and MS data) has been performed at very high standard, and appropriate tests have been performed for the statistical analysis.

Few minor points:

1. Have the Authors checked if the culture conditions for the different cell lines have somehow affected the clustering of the cells? I would comment on that in the manuscript.
2. Page 20: "...250min gradient 2% to 60%..." provide more details about the gradient. This is unlikely to be a linear gradient.
3. Page 21: Proteins with a single Ratio count were considered accurately quantified. Despite the fact that the Authors have used an accurate method to quantify proteins, since they have averaged the protein intensity from the different replicates, I would highlight in the Supplementary Table those proteins that in some experiments were quantified with single ratio count, since they might be more prone to a less accurate quantification.
4. Page 23: Explain why the 200 most abundant plasma proteins, and not for example 50 or 100, were filtered out.

Point by point review to reviewers

Summary:

We thank the reviewers for critically evaluating our manuscript and for providing helpful comments which we have used to guide our revisions. As a result, we believe our manuscript is more comprehensive and clear.

Based on the reviewers' comments we have made a new Figure 2, Figure 5, Supplementary Figure 3, and we have updated Figure 3 and Supplementary Table 1.

The new Figure 2 now includes the coverage of pathways related to cancer.

The most important addition to the revised manuscript is that we have now integrated survival data from the TCGA analysis. This shows that there is a significant survival difference between the two TCGA clusters in our study (Fig. 5c). The new Figure 5 also incorporates the recent study by Ince et al., as requested, and shows that the 67 proteins identified in our study are also differentially expressed between the Ince cluster 1 and cluster 2 cell lines (Fig. 5d).

In response to reviewer 3, we provide a new Supplementary Figure 3b, which shows that an important feature of the PCA analysis is the separation of our samples in component 2: in the integrated cell line/tumor tissue analysis, epithelial/mesenchymal protein levels drive the clustering on component 2.

The updated Supplementary Table 1 highlights single ratio counts, as requested, and in line with this, the peptide counts (razor and unique) have been incorporated into the barplot visualization in the MaxQB dataset. The user will also be able to see the number of peptides used for quantification by moving the cursor over the bar of interest.

We have also added a paragraph discussing the relevance of the group II proteins to the discussion section. We also demonstrate that our cell line signature is not dependent on any particular cell culture medium.

In addition to the reviewers' comments we addressed the editor's request for more detailed description of the used feature selection and support vector machines approach. The method section has been extended accordingly.

Once again, we thank the reviewers for their guidance in improving our manuscript.

Reviewer #1 : Ovarian Cancer

This is an interesting and well-written report. The investigators appropriately discuss the limitations of cell line research in understanding the biology of human cancers. That being said, the question remains as to the relevance of this research. For example, the discussion of a possible role for ATRA based on this cell line analysis remains questionable. Comment required.

We appreciate the reviewer's feedback. To further highlight the clinical relevance of our findings, we have now integrated the survival analysis for the two different TCGA patient subgroups in the new Fig. 5 and updated the results and discussion sections accordingly.

In respect to ATRA and ovarian cancer treatment, the benefit in the clinical setting has been controversial. Our primary goal with our experiments with was to provide a proof of principle experiment to show that proteomic datasets can be used to select appropriate cell lines for a specific functional research question. Specifically, the cell line findings show how HGSOC cell lines from either group I or group III respond differently to ATRA treatment.

Reviewer #2: Ovarian Cancer

In this manuscript, the authors undertake LC-MS/MS based proteomic analysis of 30 cell lines, mainly of ovarian cancer origin, and identify three groups and two clusters of protein expression based upon unsupervised hierarchical clustering and principal component analysis. They also identify a discriminating 67 protein signature.

Group I contains cells lines previously identified as 'likely HGSOC' in the Domcke genomic analysis, group II a mixture of lines including some CCC, some unclassifiable lines and the two Cx lines, whilst group III contains the immortalised IOSE lines, some lines felt unlikely to be HGSOC by Domcke and, crucially, at least two lines (59M and TYCNU) with TP53 mutations felt likely to be HGSOC on genomic analysis. Thus, the data suggest that HGSOC may have two proteomic signatures (exemplified by groups I and III). These data are supported by analysis of 8 primary HGSOC tumour from 5 patients. Analysis of publically available proteomic data from the TCGA sample set also supports a potential binary proteomic division of HGSOC with the 84 samples clustering into two distinct groups (TCGA-A and TCGA-B), with differences in cell of origin (fallopian tube vs ovary) postulated. The authors then suggest that differential expression of retinoic acid components could identify cell lines in groups I and III with differential sensitivity to ATRA treatment.

Overall, the work is of high quality and the MS pipeline is technically impressive. The manuscript is well written and the figures extremely clearly presented. I think that these are important data and will be of interest to the ovarian cancer community. However, there are several key questions raised by the results.

The main question relates to the ultimate utility of the proteomic signatures and how are they to be taken forward. For example, there are several clinical questions - are there prognostic implications of the two different HGSC protein clusters? Is there enough clinical information in the TCGA sample data to allow any prognosis implications to be drawn or even inferred? If it could be shown that the 67 protein

signature was prognostic in patient samples, it would increase the significance of the data.

As described above for reviewer 1, we have now explored if the signature is prognostic in the used TCGA validation dataset. Indeed, the new Fig. 5 shows that the mesenchymal ovarian cancers have an adverse prognosis.

Second, can this signature ever be robust enough or simple enough for routine clinical use? If not, is there an obvious genotype/protein phenotype relationship that could be utilised for stratification and/or prognostication?

Because our signature is based on a relatively small number of patients, we believe that confirmatory future studies will be necessary to address which of the 67 proteins in our signature can be utilized in clinical practice, e.g. to identify an optimal set of markers for immunohistochemical (IHC) profiling in combination with known markers (such as P53, PAX8). Whereas current IHC marker combinations have been proven to be successful in discriminating patients with HGSOC from other OvCa subtypes, they fail to further sub-divide patients into distinct molecular HGSOC subtypes. To our knowledge, there is currently no clinical tool available to distinguish between HGSOC subtypes. Building on the previous studies (Yang *et al.*, 2013¹; Ince *et al.*, 2015 ref²), our data now suggests a HGSOC stratification into epithelial (better prognosis) or mesenchymal (worse prognosis) HGSOC.

In addition, Ince et al (Nat. Comms. 2015) undertook gene expression and RPPA analyses of primary cell cultures and some established lines, identifying two major clusters - I cannot see reference to this manuscript here. What overlap exists between the data presented here and those previous data (especially the RPPA analyses)?

We had not compared our data to that from the Ince et al. study as there were only three cell lines in common between that study and ours. Still, as the reviewer points out, they too identified two major cell line (and patient tumor) clusters. Reviewing their data (Fig.S2a, Table S9 ref ²), we found that the 67 proteins identified by us were indeed differentially expressed between their cluster 1 or cluster 2 cell lines (**Figure R1**). Of note, expression of our group III genes was higher in their cluster 1 cell lines representative of the poor prognosis patient group (Ince et al., Fig. 6). Also expression of our group I genes was higher in their cluster 2, which represents good prognosis. These findings add important information to our manuscript and are included in the new Fig. 5d.

The results section has been updated accordingly.

Figure R1: Distribution of the 67-proteins in the Ince dataset

Specific points:

1. Figure 2b - 'frequently altered in ovarian cancer' - how was this group of proteins defined?

This group of proteins was defined through a review of the literature³, the TCGA study⁴ and the study by Domcke *et al*⁶. We have now cited the references together with the results for Fig. 2b.

2. Figure 2c - it is not quite clear what this is trying to show: I agree that there is obvious KRAS expression in KURAMOCHI, but other lines with known KRAS mutations (e.g. HeyA8) don't show up here.

To clarify this analysis, we note that Fig. 2c shows genomic amplification, and to which degree they are reflected at the protein level, rather than mutations. However, we agree that KRAS expression was not obvious in the other cell lines as depicted in the figure. To address this, we now plot log protein levels in the new Fig. 2c, allowing clearer representation of very low relative protein levels. This format is now also consistent with the relative protein levels shown in Fig. 3a.

3. Figure 2d - pathway enrichment of the proteins that are most variably expressed: again, what is this trying to demonstrate?

The aim with this analysis was to first view the data in an global fashion before moving on to a more in-depth analysis in Fig. 3. In Fig. 2d, we used the globally differentially expressed proteins from Fig. 2b. Fig. 2d highlights the underlying pathways corresponding to the most differentially expressed proteins across the different cell lines.

Please note that Fig. 2d has now become Fig. 2e.

4. In the PCA analysis in Figure 3b, it is not clear why OVCAR5 are grouped in group II not group I.

OVCAR5 is an unusual and interesting cell line; it has wildtype p53 status and while it clusters with group II in the heatmap, it seems closer to group I in the PCA in Fig. 3b. To highlight its juxtaposition between group I and group II, we have removed the lines from the PCA in Fig. 3b. In general, group II cell lines have levels of epithelial proteins, such as CDH1, similar to those of group I; this epithelial influence is represented by component 1 in the PCA. However, group assignment is based on the unsupervised hierarchical clustering from Fig. 3a which is based on the relative expression of 8,487 proteins, and this is why OVCAR5 ultimately belongs to group II.

5. Figure 4a - having 8 tumours from 5 patients strengthens the paper, but 8 is not very many, especially given that one patient (HGSOC-5) appears to behave differently from the others. What is going on with this tumour - is it possible to identify a critical difference between this tumour (e.g. type of p53 mutation, CCNE1 amplification) compared to the others? This also relates to question about the ultimate utility of these data above.

We agree that 8 tumors is not a large number. It is for this reason that we used the publically available TCGA proteomic data to validate the signature established with the cell lines. The two TCGA clusters represent a predominantly epithelial FTEC-derived HGSOC subtype and a mesenchymal OSE-derived HGSOC subtype. A few proteins (ITGA5, HMOX1, GJA1, SACS, SMTN) most likely represent specific markers for the mesenchymal HGSOC subtype. These proteins are highly expressed in the TCGA-B cluster and in the HGSOC-5 tumor. Accordingly, it appears that the critical difference is that HGSOC5 belongs to the mesenchymal, OSE-derived HGSOC subtype.

Furthermore, our comparison to the Ince et al study now provides further independent validation of our results.

6. I regret that Figure 6b is not terribly convincing.

The goal of Fig. 6b was to perform pairwise comparisons to identify the strongest enriched pathways in the different groups and to show the results of the pairwise comparison in a volcano plot format. One of the most interesting findings is that five Vitamin A pathway proteins are highly expressed in group I cell lines. Although the position of some of the proteins in this pathway may not appear significant in this plot, there was a statistically very significant difference for the Vitamin A annotation (Benjamini-Hochberg false discovery rate $p = 0.003$). We believe that Fig. 6b is an important figure, providing the reader with visual confirmation of the enrichment of the whole Vitamin A pathway in group I cell lines.

7. Culture medium. What effect does culture medium have upon protein expression? Ince et al also demonstrated that the nature of culture medium could have profound effects upon growth and even copy number profiles of cells that grew from primary cultures. Have the authors demonstrated the reproducibility of signatures in different culture media?

The new Supplementary Table 4 lists the growth media for each cell line. While group II cell lines were all cultured in DMEM, the culture media for group I and group III cell lines, which both contain HGSOC cell lines, consisted of different media, suggesting that the signature is not dependent on any particular culture medium. A column has been added to the new Supplementary Table 4, identifying the group each cell line belongs to, based on our proteomic analysis. This shows that the grouping and signature is not cell culture-dependent. To further clarify this,

Figure R2: Inclusion of media in Fig. 3b heatmap

Figure R2 integrates the heatmap from Figure 3a and the different cell lines media so that the cell lines can be evaluated together with their respective media (**Figure R2**). In the revised manuscript, we now note this finding in the discussion section, page 14.

Minor point

1. Supplementary table 3 is labelled as supplementary table 4.

This was corrected.

Reviewer #3: tumour microenvironment and proteomics

In this manuscript, Coscia F. and colleagues have used single-run mass spectrometry to perform in-depth proteomic analysis of 26 ovarian cancer cell lines commonly used for experiments, 2 cervical cancer, 2 immortalised ovarian surface epithelial cells, three primary fallopian tube epithelial cell isolates and 7 ovarian high grade ovarian cancer tissues. By using statistics and bioinformatics, the Authors have identified three separated clusters of cell lines with different proteomic signatures and using supporting vector machine they have defined a 67-protein cell line signature which separated cell lines into epithelial (containing fallopian tube epithelial cell) and mesenchymal (containing ovarian surface epithelial cells) type, and which was able to similarly segregate patient samples data from CPTAC/TCGA tumour proteome dataset. The Authors hypothesised that their signature can stratify patient samples according to the origin of the ovarian cancer. Moreover, the Authors provide confirmation of CREBP2 levels in the different cell lines using western blot, and CREBP2 expression in high grade serous ovarian cancers using IHC on patient samples. Furthermore, they provide functional validation of retinoic acid pathway regulation using ATRA treatment on ovarian cancer cells.

This is an original work which uses an unbiased and state-of-the-art, highly accurate quantitative approach to answer key open questions in the ovarian cancer fields. The Authors provide first evidence that proteomics has the potential to successfully identify the origin of ovarian cancers. Moreover, they provide the proteomic analysis of almost 30 commonly used ovarian cancer cell lines as valuable resource, which, combined to previously published genomic analysis, will help researchers to select the appropriate cancer cell model for their study. I strongly believe that the robustness of this work and the findings make this manuscript suitable for Nature Communications. However, there are several issues that need to be addressed by the Authors.

1. Page 5: The Authors claim that for this work they have developed a method based on single-run described recently. What is exactly the development that they have made and how has this improved the previous method?

We agree that the single-run method is not a new development. This paper is the first application of the original concept⁶ to cell lines, primary cells and tissues, with equally good results. To achieve this, we had to adapt the up-front sample preparation as well as make use of the ongoing improvements in MaxQuant analysis. This point is now clarified in the Results section.

2. Page 6 "protein expression values": expression should be replaced by levels,

since protein levels do not necessarily depend only on expression, but also degradation and other mechanisms.

This was changed as suggested.

3. Fig S1d: The coverage of pathways related to cancer is interesting information. I would move that panel in the Figure 1 of the manuscript.

Re-reading the manuscript with the benefit of time, we agree with the reviewer that it should be in a main figure. It appears to us that moving Fig. S1d to Figure 2d, directly before the pathway enrichment analysis contributes to the logical progression of the manuscript.

4. Page 7 and Fig 2b: the Authors should explain what iBAQ has been used in the plot. Is that the sum of the iBAQ calculated by MaxQuant for the cell lines?

This was changed to “sum of the intensity-based absolute quantification (iBAQ) values calculated by MaxQuant” on Page 7 and in the legend for Figure 2b.

5. Fig 2c. To make their statement more solid, the Authors should expand the range of examples and show a broader panel of known deleted, amplified and mutated genes in ovarian cancer cell lines (e.g. based on Ref 9), including for example Myc, Rb1 and p53.

In Figure 2c the focus is on genomic amplification, rather than mutations, as amplifications are more likely to be reflected at the protein level. Still, p53 mutations play an important role in ovarian cancer and we have investigated the association of p53 mutation with protein levels across the 30 cell lines (Fig. 3a). Here we see that p53 or wildtype mutation status does not correlate with relative p53 protein levels. This may be explained by the different p53 mutations in the various cell lines, some of which may affect protein levels, while others may, for example, inactivate the protein. For this reason, we believe that p53 information is best represented in Fig. 3a, along with the relative levels of other proteins that are frequently altered in ovarian cancer, rather than in Fig. 2c.

Based on the reviewer’s suggestion, we have investigated the protein expression levels of both Myc and Rb1. Our analysis detected only very low levels of MYC, making it unsuitable for inclusion in Fig. 2c. The levels of the important oncoprotein Rb1 did not vary significantly across cell lines and it is included in Fig. 3a. Figure 3a also includes a number of additional proteins that are frequently altered in OvCa.

6. Page 9: proteins mentioned as examples for group II and III seem to have been chosen somehow randomly. Could the Authors rationally explain why they chose them as examples?

Much of **group I** consisted of cell lines that were previously reported to likely represent HGSOC cell lines, based on features of their genomic profiles⁵. This was also reflected in their proteomic profile as a number of known HGSOC markers were highly expressed in this group. We therefore selected three of these known HGSOC markers as examples for group I (PAX8, MSLN, MUC16). We now state that selection of these proteins was based on our review of the literature. In particular,

- Immunohistochemical staining for **PAX8** is used in the differential diagnosis of HGSOC and it is a marker of the tubal epithelium⁷.
- **MSLN** is a marker for detecting ovarian epithelial cancers⁸, and it is known to promote the migration and invasion of OvCa cells⁹.
- **MUC16** (CA-125) is elevated in 80% of advanced stage OvCa and is the only tumor biomarker recommended for clinical use in OvCa diagnosis and management.

Group II is the least HGSOC-like group of our three cell line groups and it contained the OVISE clear cell cancer cell line. The clear-cell subtype of ovarian cancers displays a gene signature that easily distinguishes this subtype from the other OvCa subtypes; this signature includes AKR1C1¹⁰, a protein whose expression was highest in group II. Based on this apparent clear cell cancer influence in group II cell lines, we chose to highlight another clear cell cancer marker (HNF1B).

- Gene expression profiling of ovarian carcinomas have identified **HNF1B** to be amongst the most upregulated transcripts in ovarian clear cell cancer compared to the other histological subtypes¹¹.

While containing lower levels of the well-known epithelial markers of HGSOC described for group I, **group III** cells contained higher levels of the known mesenchymal markers of HGSOC (ITGA5, VIM, FN1). HMOX1, a protein that has not been characterized in OvCa was one of the driver proteins in this group. Therefore, we selected to highlight these four proteins:

- **ITGA5** is one of the driver proteins in group III, and has been shown to mediate early OvCa metastasis¹²
- MMP-2 cleavage of **FN1** mediates the initial steps of ovarian cancer metastasis¹³
- **VIM** and **FN1** are overexpressed in the process of epithelial-to-mesenchymal transition¹⁴
- **HMOX1** is one of the driver proteins in group III. It has been shown to contribute to cisplatin-resistance in lung cancer and OvCa cell lines¹⁵, in line with its expression in the more mesenchymal tumors in our study, which are likely to be chemoresistant.

We have also provided further information on other proteins of interest in Supplementary Table 3.

7. Fig 4b and page 10. The Authors comment extensively on component 1. However, component 2 suggests that there are some proteomic similarities between HGSOC and IOSEs in Group III. Please comment on this as well. Considering the Authors' findings later in the manuscript, could component 1 represent cell proliferation?

We thank the reviewer for bringing up this very interesting point, which turned out to be a very useful addition to the manuscript. We have now looked at the proteins driving the separation in both dimension and performed a pathway enrichment analysis. Indeed, the DNA replication pathway is main contributor to component 1 separation with a clear tendency to be higher in samples positioned on the right side of the PCA. Figure 6a shows the results of the annotation enrichment analysis of all the cell lines with pathways representing component 1 factors.

The new analysis of component 2 revealed that it represents differences in epithelial/mesenchymal protein levels. Lower on component #2 (FTECs), there is higher expression of epithelial proteins such as EPCAM, CDH1, KRT7, FOLR1 and PAX8; higher

on the component 2 axis, the levels of mesenchymal proteins increase and those of the epithelial proteins decrease. Therefore, the top of component 2 represents the mesenchymal features of the IOSEs, group III, and the tumors. We have generated a new supplementary figure of this important observation (Supplementary Fig. 3b), showing the proteins that drive the segregation.

8. Page 10. The discussion of ITGA5 and AKR1C1 is not clear.

Are the Authors trying to say that the three groups could somehow represent three different ovarian cancer types? Information about ITGA5 and AKR1C1 could have been mentioned already at page 9, and here the Authors could just highlight that those proteins are included in the 67 protein signature.

We would like to note that the three groups do not definitively represent three different ovarian cancer histologies. The clustering is due to the differential levels of epithelial, mesenchymal, and clear cell cancer proteins. Although group I likely contains all HGSOC cell lines, group III is a mixture of known HGSOC, endometrioid, and previously uncategorized cell lines. Similarly, group II is comprised of a number of different types of cell lines, including the clear cell cancer line OVISe, and the hypermutated IGROV1 cell line described in ref ⁵.

With respect to the ITGA5 and AKR1C1, we have updated the section on Page 9 accordingly.

9. Fig 4e: S4a and Page 11. The Authors should write somewhere (Methods or Figure legend) how many tissues have been stained to conclude that CRABP2 is not expressed in HGSOC but not normal OSE and FTEC. Fig S4b: To strengthen their conclusion, in addition to confirming the proteomic data for CRABP2 by western blot, the Authors should provide CRABP2 staining of non-HGS ovarian cancers, such as clear cell and endometrioid ovarian cancers, for comparison to HGSOC.

Figure R3: CRABP2 in Oncomine database for comparison to HGSOC.

The Methods section has been updated to include the number of tissues stained.

With respect to CRABP2 expression in HGS and non-HGS ovarian cancers, our discussion references previous studies that have reported that CRABP2 is higher in the serous subtype of ovarian cancer compared with the clear cell, endometrioid, and mucinous subtypes¹⁶. Additionally, the Oncomine database confirmed that CRABP2 is also higher in the HGS subtype (Schwartz ovarian cancer dataset¹⁷) (**Figure R3**). Taken together this is strong evidence that CRABP2 is higher in the serous ovarian cancer.

10. Page 11: The Authors mention only here for the first time the presence of mesenchymal markers in group III. I would mention it earlier on, when the Authors comment about group I, II and III. Page 12. Provide references for ITGA5, HMOX1, SMTN and GJA1 to be mesenchymal markers.

The results section has been updated accordingly.

11. Page 11-13 and Figs 5 and 6: I find confusing that in the paragraph "Integrative comparison of HGSOC..." the Authors compare Cluster 1 (= group I) with Cluster 2 (=group II and III), then in the following paragraph "HGSOC from TCGA..." they consider group I and group III, and then in paragraph "Utility of the proteomics..." they go back again to compare Cluster 1 with Cluster 2. I think that the comparison between the two clusters (last paragraph) gives a clear rationale on why to focus on CRABP2 for validation, rather than mentioning (page 11) "we first focused on CRABP2 because retinoic acid pathway was differentially expressed..."

Indeed, many other processes were differentially regulated. After the validation of CRABP2, the Authors could do functional validation with ATRA. Then, they could finish with the utility of the 67 protein signature to group patients cancer tissues.

In the TCGA results section, the focus is on group I and group III proteins and their respective associations with the TCGA-A and TCGA-B sub-groups; group II proteins were

not detected in the TCGA. This is not surprising because the TCGA tumors are HGSOE, and many group II proteins belong to the clear cell subtype. In the analysis in the previous paragraph, “cluster 1” contained group I cell lines, and “cluster 2” contained both group II and group III cell lines. Due to the absence of group II proteins in the TCGA, using the term “cluster 2” in this section would have been inaccurate.

With respect to the location of the TCGA and ATRA paragraphs, we would like to retain the current format. In the resubmitted manuscript, the 67-protein signature is now shown to be relevant in the clinical setting before the functionality of specific proteins is investigated.

12. Page 11-13: The relevance/meaning of Group II is not clear. The Author should discuss more clearly what they think this cluster represents. For example, looking at figure 3c, Group II cell lines seem more epithelial-like, since they have high levels of CDH1. However, they have similar levels of ITGA5 compared to group III, which is considered more mesenchymal. The Authors should address this point.

A new paragraph discussing the relevance of group II has been included in the discussion section.

13. Page 12: The Authors wrote that Cluster 2 contains higher levels of proteins involved in mitosis. Have the Authors checked, or are data already available, the proliferation of these cell lines? Could it be that proliferation is major discriminating factor between Cluster 1 and Cluster 2 rather than the epithelial/mesenchymal status of the cells?

Yes, proliferation-associated proteins are higher in cluster 2. This increased proliferation is in line with a study by Beaufort et al. that evaluated the doubling times of 39 OvCa cell lines and reported on the higher proliferative capacity of putative endometrioid and CCC cell lines compared with putative HGSOE cell lines¹⁸. This study is referenced in our results section. As this was not a novel finding, we focused on the epithelial/mesenchymal component of our study, which was both novel and a relevant feature of our cell line and tumor clusters.

14. Page 12-13: to strengthen their finding the Authors should perform the ATRA experiment in CRABP2 silenced cells, to show the specificity of the measured proliferation effect. Alternatively they should use a second approach, similar to ATRA.

The primary aim of the ATRA treatment was a proof of principle experiment showing that the proteomic dataset can be used to select appropriate cell lines for a specific question. Given the other components in the Vitamin A pathway, it is possible that CRABP2-silencing experiments may not alter the effect of ATRA. Our goal was simply to demonstrate that our proteomic resource would allow researchers to work with the optimal cell lines for specific questions.

15. Page 11. Typo: Supplementary Fig 4a and not 8a.

This has been updated accordingly.

Data analysis:

Overall, the proteomic analysis (MS samples and MS data) has been performed at very high standard, and appropriate tests have been performed for the statistical analysis.

Few minor points:

1. Have the Authors checked if the culture conditions for the different cell lines have somehow affected the clustering of the cells? I would comment on that in the manuscript.

As noted above for reviewer 2, we have now added an extra column to the new Supplementary Table 4, identifying the group each cell line belongs to. This shows that the grouping and signatures are not cell culture-dependent. Additionally, the heatmap from Figure 3b was redone so that the grouping of the cell lines can be evaluated together with their respective media (**Figure R2**).

2. Page 20: "...250min gradient 2% to 60%..." provide more details about the gradient. This is unlikely to be a linear gradient.

We have now provided HPLC gradient details in the corresponding method section.

3. Page 21: Proteins with a single Ratio count were considered accurately quantified. Despite the fact that the Authors have used an accurate method to quantify proteins, since they have averaged the protein intensity from the different replicates, I would highlight in the Supplementary Table those proteins that in some experiments were quantified with single ratio count, since they might be more prone to a less accurate quantification.

We agree that providing more such details will be helpful to evaluate quantification accuracy and to interpret our data. We therefore highlighted single ratio counts in red in the Supplementary Table 1. Furthermore, we have also integrated peptide counts (razor + unique) into the barplot visualization to compare protein levels across cell lines in our MaxQB database. Bar widths represent the number of quantified peptides per sample in relation to the maximum number of possible tryptic peptides. The wider the bar, the more peptides were present. The user can also see the number of peptides used for quantification by directly moving the mouse cursor on a bar of choice. We attach a screenshot as an example (**Figure R4**). The dataset can be accessed on the MaxQB website (<http://maxqb.biochem.mpg.de/mxldb/project/show/P017>, User: review5, Password: 3t6tyC) and will be released for open access after publication.

Experiment Series Ovarian Cancer Models

Figure R4. Screenshot of example MaxQB output

4. Page 23: Explain why the 200 most abundant plasma proteins, and not for example 50 or 100, were filtered out.

The total list of 243 (roughly 200) plasma proteins was derived from a recently published paper by our group¹⁹. These proteins are frequently found in single-run proteomic analysis of plasma samples across individuals and make up 98% of the total plasma protein mass when compared to a deep fractionated plasma proteome measured to a depth of 1492 proteins. The figures below summarize this fact.

Deep fractionated plasma proteome
(total 1492 proteins, Geyer et al., 2016)

REFERENCES

- 1 Yang, D. *et al.* Integrated analyses identify a master microRNA regulatory network for the mesenchymal subtype in serous ovarian cancer. *Cancer Cell* **23**, 186-199 (2013).
- 2 Ince, T. A. *et al.* Characterization of twenty-five ovarian tumour cell lines that phenocopy primary tumours. *Nature communications* **6**, 7419 (2015).
- 3 Ross, J. S. *et al.* Comprehensive genomic profiling of epithelial ovarian cancer by next generation sequencing-based diagnostic assay reveals new routes to targeted therapies. *Gynecol Oncol* **130**, 554-559 (2013).
- 4 Cancer Genome Atlas Research Network. Integrated genomic analyses of ovarian carcinoma. *Nature* **474**, 609-615 (2011).
- 5 Domcke, S., Sinha, R., Levine, D. A., Sander, C. & Schultz, N. Evaluating cell lines as tumour models by comparison of genomic profiles. *Nature communications* **4**, 2126 (2013).
- 6 Thakur, S. S. *et al.* Deep and highly sensitive proteome coverage by LC-MS/MS without prefractionation. *Mol Cell Proteomics* **10**, M110 003699 (2011).
- 7 Bowen, N. J. *et al.* Emerging roles for PAX8 in ovarian cancer and endosalpingeal development. *Gynecol Oncol* **104**, 331-337 (2007).
- 8 Huang, C.-Y. *et al.* Serum mesothelin in epithelial ovarian carcinoma: a new screening marker and prognostic factor. *Anticancer research* **26**, 4721-4728 (2006).
- 9 Chang, M.-C. *et al.* Mesothelin enhances invasion of ovarian cancer by inducing MMP-7 through MAPK/ERK and JNK pathways. *The Biochemical journal* **442**, 293-302, doi:10.1042/bj20110282 (2012).
- 10 Schaner, M. E. *et al.* Gene expression patterns in ovarian carcinomas. *Molecular biology of the cell* **14**, 4376-4386 (2003).
- 11 Tsuchiya, A. *et al.* Expression profiling in ovarian clear cell carcinoma: identification of hepatocyte nuclear factor-1 beta as a molecular marker and a possible molecular target for therapy of ovarian clear cell carcinoma. *Am J Pathol* **163**, 2503-2512 (2003).
- 12 Sawada, K. *et al.* Loss of E-cadherin promotes ovarian cancer metastasis via alpha 5-integrin, which is a therapeutic target. *Cancer Res* **68**, 2329-2339 (2008).
- 13 Kenny, H. A., Kaur, S., Coussens, L. M. & Lengyel, E. The initial steps of ovarian cancer cell metastasis are mediated by MMP-2 cleavage of vitronectin and fibronectin. *J Clin Invest* **118**, 1367-1379, doi:10.1172/JCI33775 (2008).

- 14 Kalluri, R. & Weinberg, R. A. The basics of epithelial-mesenchymal transition. *J Clin Invest* **119**, 1420-1428, doi:10.1172/JCI39104 (2009).
- 15 Song, J., Shih, I.-m., Chan, D. W. & Zhang, Z. Suppression of annexin A11 in ovarian cancer: implications in chemoresistance. *Neoplasia* **11**, 605-614, 601 p following 614 (2009).
- 16 Toyama, A. *et al.* Proteomic characterization of ovarian cancers identifying annexin-A4, phosphoserine aminotransferase, cellular retinoic acid-binding protein 2, and serpin B5 as histology-specific biomarkers. *Cancer science* **103**, 747-755 (2012).
- 17 Schwartz, D. R. *et al.* Gene expression in ovarian cancer reflects both morphology and biological behavior, distinguishing clear cell from other poor-prognosis ovarian carcinomas. *Cancer Res* **62**, 4722-4729 (2002).
- 18 Beaufort, C. M. *et al.* Ovarian Cancer Cell Line Panel (OCCP): clinical importance of in vitro morphological subtypes. *PLoS One* **9**, e103988 (2014).
- 19 Geyer, Philipp E. *et al.* Plasma proteome profiling to assess human health and disease. *Cell Systems* **2**, 185-195 (2016).

REVIEWERS' COMMENTS:

Reviewer #2 (Remarks to the Author):

As stated before, this is an impressive piece of research that will be significant value to the ovarian cancer research community.

The authors have updated this manuscript to address the comments from all three reviewers. For this reviewer, I think that the inclusion of survival data from TCAG in new figure 5C and the correlation with results from previous cell line analyses is important and strengthens the results.

Reviewer #3 (Remarks to the Author):

This reviewer is satisfied with the revisions provided by the Authors. The revised manuscript is a valuable Resource and suitable for publication in Nature Communications.

Reviewer #4 (Remarks to the Author):

In my view the authors have addressed the concerns of the reviewers. My only concern is that the data analysis and statistical analysis is written in a way that reproducing the results would be hard. Stating the relevant equation for MaxQuant and providing the data analysis scripts would address this.

Response to Reviewers

Once again, we thank the reviewers for their comments and guidance, which have improved the content of our manuscript. Below we address the final requests from reviewer 4.

REVIEWERS' COMMENTS:

Reviewer #2 (Remarks to the Author):

As stated before, this is an impressive piece of research that will be significant value to the ovarian cancer research community.

The authors have updated this manuscript to address the comments from all three reviewers. For this reviewer, I think that the inclusion of survival data from TCAG in new figure 5C and the correlation with results from previous cell line analyses is important and strengthens the results.

Reviewer #3 (Remarks to the Author):

This reviewer is satisfied with the revisions provided by the Authors. The revised manuscript is a valuable Resource and suitable for publication in Nature Communications.

Reviewer #4 (Remarks to the Author):

In my view the authors have addressed the concerns of the reviewers. My only concern is that the data analysis and statistical analysis is written in a way that reproducing the results would be hard. Stating the relevant equation for MaxQuant and providing the data analysis scripts would address this.

Response: The freely available Perseus software which we used to analyze the data is publically available. The manuscript, which we supplied for the revision, will appear in one week, on June 28th, in time for referencing it in the galley stage (temporary reference already included). We have supplied a paragraph regarding data availability in our manuscript for access to the proteomic raw files. Furthermore, the authors are happy to answer any questions regarding the data analysis in Perseus that users may have and can be contacted by email.